# Multiomics study of nonalcoholic fatty liver disease

Gardar Sveinbjornsson [1,20] ✉, Magnus O. Ulfarsson[1,2,20], Rosa B. Thorolfsdottir[1], Benedikt A. Jonsson[1], Eythor Einarsson[1], Gylfi Gunnlaugsson[1], Solvi Rognvaldsson[1], David O. Arnar[1,3,4], Magnus Baldvinsson[5], Ragnar G. Bjarnason[3,6], DBDS Genomic consortium*, Thjodbjorg Eiriksdottir [1], Christian Erikstrup [7], Egil Ferkingstad[1], Gisli H. Halldorsson[1], Hannes Helgason[1], Anna Helgadottir [1], Lotte Hindhede [7], Grimur Hjorleifsson [1], David Jones[8], Kirk U. Knowlton[9], Sigrun H. Lund [1], Pall Melsted[1,10], Kristjan Norland[1], Isleifur Olafsson[11], Sigurdur Olafsson[4], Gudjon R. Oskarsson [1], Sisse Rye Ostrowski [12,13], Ole Birger Pedersen [13,14], Auðunn S. Snaebjarnarson [1], Emil Sigurdsson[15,16], Valgerdur Steinthorsdottir [1], Michael Schwinn[12], Gudmundur Thorgeirsson[1,4], Gudmar Thorleifsson [1], Ingileif Jonsdottir [1,3], Henning Bundgaard[13,17], Lincoln Nadauld[8], Einar S. Bjornsson[3,4], Ingrid C. Rulifson[18], Thorunn Rafnar [1], Gudmundur L. Norddahl[1], Unnur Thorsteinsdottir[1,3], Patrick Sulem[1], Daniel F. Gudbjartsson[1,2,19], Hilma Holm [1] and Kari Stefansson [1,3] ✉

Nonalcoholic fatty liver (NAFL) and its sequelae are growing health problems. We performed a genome-wide association study of NAFL, cirrhosis and hepatocellular carcinoma, and integrated the findings with expression and proteomic data. For NAFL, we utilized 9,491 clinical cases and proton density fat fraction extracted from 36,116 liver magnetic resonance images. We identified 18 sequence variants associated with NAFL and 4 with cirrhosis, and found rare, protective, predicted loss-of-function variants in *MTARC1* and *GPAM*, underscoring them as potential drug targets. We leveraged messenger RNA expression, splicing and predicted coding effects to identify 16 putative causal genes, of which many are implicated in lipid metabolism. We analyzed levels of 4,907 plasma proteins in 35,559 Icelanders and 1,459 proteins in 47,151 UK Biobank participants, identifying multiple proteins involved in disease pathogenesis. We show that proteomics can discriminate between NAFL and cirrhosis. The present study provides insights into the development of noninvasive evaluation of NAFL and new therapeutic options.

NAFL disease (NAFLD), when >5% of the liver is fat with no identifiable secondary cause, is estimated to affect 25% of the world's population[1–3]. NAFL, the first stage of NAFLD, can progress to nonalcoholic steatohepatitis (NASH)[4,5]. Furthermore, some patients with NASH develop liver cirrhosis and hepatocellular carcinoma (HCC). Obesity, metabolic syndrome, diabetes and hypertension are recognized risk factors for NAFL and NASH[1,6,7], and NASH-related cirrhosis is the second most common indication for liver transplantation in the United States of

A full list of affiliations appears at the end of the paper. *A list of members and their affiliations appears in the Supplementary information.
✉e-mail: gardars@decode.is; kstefans@decode.is

America[1,8,9]. Several sequence variants have been associated with liver enzymes[10–12], cirrhosis[13] and NAFLD[14–16], including missense variants in *PNPLA3* (ref. [17]), *TM6SF2* (ref. [18]), *GCKR* (ref. [19]) and *MTARC1* (ref. [20]). Furthermore, a protein-truncating variant in *HSD17B13* has been associated with NASH and fibrosis but not with steatosis[21].

It is challenging to both diagnose and stage NAFLD. Although liver enzymes are commonly elevated, they are nonspecific and poor predictors of progression[22–24]. Magnetic resonance imaging (MRI)-derived proton density fat fraction (PDFF) provides accurate liver fat quantification, but liver biopsy is essential for NASH diagnosis and staging. However, biopsy involves sampling variability and a risk of complications[3,25]. Currently, no pharmacological therapy has been approved for NASH. The identification of potential drug targets and biomarkers to monitor disease progression and treatment response is therefore paramount.

We performed genome-wide association studies (GWASs) of PDFF, NAFL, cirrhosis and HCC, and a combined GWAS of PDFF and NAFL. To further characterize the risk variants, we tested them for association with clinical traits, mRNA expression in various tissues and circulating protein levels in large datasets from Iceland and the UK Biobank (UKB). We further investigated whether plasma proteins discriminate between disease stages.

## Results

### GWAS analysis

We extracted liver PDFF from raw abdominal MR images of 36,116 Britons of European ancestry from the UKB (Fig. 1, Table 1 and Methods). The MR images were generated using two acquisition techniques, that is, 8,448 images using gradient multiecho (GRE) and 27,668 using iterative decomposition of water and fat with echo asymmetry and least-squares estimation (IDEAL). The results of the two methods were similar, with 20.0% of IDEAL images and 23.6% of GRE images having a PDFF > 5%. Of the 270 (IDEAL) and 97 (GRE) individuals diagnosed with NAFL, 179 (66.3%) and 76 (78.4%) had a PDFF > 5.0% (Supplementary Fig. 1). Our PDFF estimates correlate well ($r^2 = 0.96$) with 3,869 PDFF measurements calculated by others using LiverMultiScan[26] (Supplementary Fig. 2).

We performed a GWAS on the PDFF estimates ($n = 36,116$) using each individual's first available MRI measurement. We also meta-analyzed GWASs on clinically diagnosed NAFL (*International Classification of Disease*, 10th revision (ICD-10)[27], code K76.0), including 785 cases from Iceland (deCODE genetics), 5,921 from the UK (UKB), 2,134 from the USA (Intermountain INSPIRE and HerediGene registries) and 651 from Finland (FinnGen), for a total of 9,491 NAFL cases and 876,210 controls. We combined the summary-level GWAS PDFF data and the NAFL ICD-10 code meta-analysis data to maximize power to detect associations using multitrait analysis[28].

We identified 18 independent sequence variants at 17 loci in the combined GWAS (Table 2 and Supplementary Table 1), of which 4 have not been reported in an NAFL GWAS (in/near *PNPLA2*, *TOR1B*, *APOH* and *GUSB*). The variants associated with the combined PDFF and NAFL at genome-wide significance (GWS) were nominally significant for both phenotypes ($P < 0.05$). Furthermore, their effects on the two phenotypes were comparable (Fig. 2), suggesting that variants that increase PDFF also increase risk of NAFL and vice versa. We therefore refer to the combined phenotype as NAFL. The strongest association was with missense variant p.Ile148Met in *PNPLA3* ($P = 3.0 \times 10^{-217}$, effect = 0.28 s.d. for PDFF and $P = 9.7 \times 10^{-116}$, odds ratio (OR) = 1.47 for NAFL). This variant has been reported as being associated with NAFLD[17,29]. Two of the eighteen variants had greater effect on PDFF in men than in women, p.Ile43Val in *GPAM* (effect in men = 0.09 s.d., effect in women = 0.04 s.d., $P_{\text{males versus females}} = 0.00099$) and p.Glu167Lys in *TM6SF2* (effect in men = 0.40 s.d., effect in women = 0.28 s.d., $P_{\text{males versus females}} = 7.3 \times 10^{-5}$).

The NAFL variants include a low-frequency (minor allele frequency (MAF) = 2.96%) missense variant, p.Cys325Gly, in *APOH*

(encoding β$_2$-glycoprotein 1) ($P = 4.0 \times 10^{-9}$, effect = 0.13 s.d. for PDFF and $P = 0.02$, OR = 1.11 for NAFL). *APOH* is highly expressed in the liver[30,31] and p.Cys325Gly has been associated with liver enzymes[16]. A low-frequency (MAF = 1.32%) missense variant, p.Asn252Lys, in *PNPLA2*, the closest homolog of *PNPLA3* (ref. [32]), is also associated with NAFL ($P = 4.9 \times 10^{-12}$, effect = 0.22 s.d. for PDFF and $P = 0.0013$, OR = 1.16 for NAFL). Homozygous mutations in *PNPLA2* have been associated with neutral lipid storage disease and fatty liver is among its features[33]. *PNPLA2* p.Asn252Lys has been associated with increased waist:hip ratio (WHR) and high-density lipoprotein-cholesterol levels[34,35] but not with NAFL. Adjusting for the WHR does not affect the association ($P = 2.4 \times 10^{-10}$, effect = 0.20 s.d. for PDFF). A common NAFL-associated variant, rs6955582[A], intronic in *GUSB* ($P = 4.7 \times 10^{-7}$, effect = −0.038 s.d. for PDFF and $P = 0.00019$, OR = 0.95 for NAFL) correlates with a missense variant in *GUSB*, p.Leu649Pro ($r^2 = 0.87$ in the UK and 0.99 in Iceland, MAF = 44.9%). An intronic variant (MAF = 9.23%) in *TOR1B*, rs7029757[A] ($P = 9.2 \times 10^{-10}$, effect = −0.078 s.d. for PDFF and $P = 0.00028$, OR = 0.92 for NAFL) is also associated with NAFL. The rs7029757[A] has been reported as associating with alanine aminotransferase (ALT) levels and cirrhosis but not NAFL[13].

We also performed meta-GWASs on 4,809 all-cause cirrhosis (cirrhosis for simplification; Methods) cases ($n_{\text{cases UK}} = 2,301$, $n_{\text{cases Iceland}} = 691$, $n_{\text{cases USA}} = 392$, $n_{\text{cases Finland}} = 1,425$, $n_{\text{controls}} = 967,898$) and 861 HCC cases ($n_{\text{cases UK}} = 374$, $n_{\text{cases Iceland}} = 406$, $n_{\text{cases USA}} = 81$, $n_{\text{controls}} = 819,551$) using ICD-10 codes. Four variants were associated with cirrhosis (Table 3), two of which associated with PDFF (Supplementary Table 2). The two that did not associate with PDFF ($P > 0.19$) are a splice-region variant in *HSD17B13*, rs72613567[TA], which has been reported as being associated with NASH[21], and a missense variant p.Glu366Lys in *SERPINA1*, which causes α$_1$-antitrypsin deficiency[36].

We compared the PDFF and cirrhosis effects of the 18 NAFL variants in the UKB data. The cirrhosis effects were proportional to the PDFF effects (Fig. 2) except for p.His48Arg in *ADH1B* (alcohol dehydrogenase 1b) and p.Cys282Tyr in *HFE* (homeostatic iron regulator). These two variants are associated with cirrhosis through alcohol consumption (*ADH1B*)[37] and hemochromatosis (*HFE*)[38], rather than solely through hepatic fat. Similarly, the effects of the 18 variants on HCC were proportional to the PDFF effects (Fig. 2).

### Potential causal candidate genes

To prioritize plausible causal genes at the associated loci, we evaluated the lead affected amino acid sequence, mRNA expression (expression quantitative trait loci (eQTLs)) or splicing QTLs (sQTLs). For this analysis, we used annotation of 46.5 million sequence variants tested in the GWAS and measured mRNA levels using inhouse RNA-sequencing (RNA-seq) data from whole blood ($n = 17,846$) and adipose tissue ($n = 770$), and publicly available data in the Genotype-Tissue Expression project (GTEx, v.v8). Fourteen lead associations were with missense variants or variants in high linkage disequilibrium (LD; $r^2 > 0.8$) with a missense variant: in *PNPLA3*, *APOE*, *GCKR*, *GPAM*, *PNPLA2*, *TMC4*, *MTARC1*, *APOH*, *ADH1B*, *HFE*, *ERLIN1*, *GUSB* and two independent variants in *TM6SF2*. The missense variants in *GUSB*, *TM6SF2*, *PNPLA3* and *TMC4* are associated with expression levels of the corresponding gene (*cis*-eQTLs) in various tissues, and the missense variant in *TMC4* is also associated with liver expression of *MBOAT7*. Loss of *MBOAT7* has been associated with NAFLD[39–41]. These variants are top eQTLs, that is, they are either the strongest association at their loci with expression levels of these genes or highly correlated to it ($r^2 > 0.8$). The variants in *TOR1B*, *HSD17B13* and *GUSB* associate with splicing in whole blood as top sQTLs (Supplementary Table 3). The intronic variant rs7029757[A], located 50-bp downstream of exon 2 in *TOR1B* (torsin 1b), is associated with cryptic splicing ($P = 1.0 \times 10^{-1493}$; Supplementary Table 3). This variant elongates exon 2 by 50 bp, leading to a frameshift introducing a premature stop codon in exon 3 (Supplementary Fig. 3).

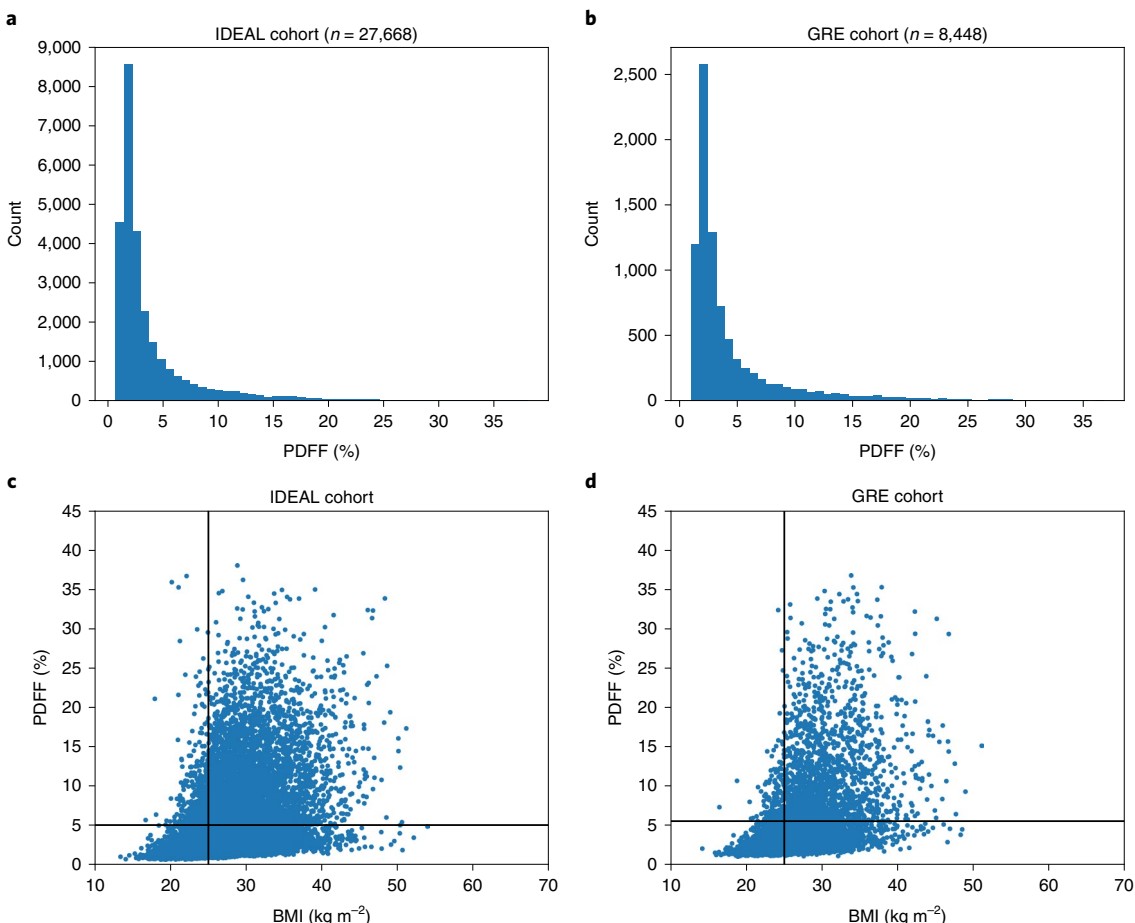

**Fig. 1 | The distribution of PDFF measurements. a**, Histogram of PDFF in the IDEAL cohort. **b**, Histogram of PDFF in the GRE cohort. **c**, PDFF plotted against BMI in the UKB IDEAL cohort ($n$ = 27,668). **d**, PDFF plotted against BMI in the UKB GRE cohort ($n$ = 8,448). The vertical line is at BMI = 25 kg m$^{-2}$. The horizontal line is at PDFF = 5%.

**Table 1 | Fraction of participants with high versus low BMI and PDFF in the UKB cohorts**

| | Low BMI, high PDFF (%) | Low BMI, low PDFF (%) | High BMI, high PDFF (%) | High BMI, low PDFF (%) |
|---|---|---|---|---|
| **IDEAL cohort** | 2.03 | 39.18 | 18.06 | 40.73 |
| **GRE cohort** | 1.87 | 37.21 | 19.24 | 41.64 |

Low BMI < 25 kg m$^{-2}$; low PDFF < 5%.

### Association of NAFL variants with other traits and outcomes

We tested the 20 NAFL and cirrhosis variants for association with 51 phenotypes related to liver function or NAFLD, using data from Iceland, the UK, Denmark (CHB-CVDC/DBDS) and available meta-analyses (Supplementary Table 4)[42–45]. The risk alleles of all variants were associated with increased levels of at least one liver enzyme (Figs. 2 and 3). Most variants were associated with cholesterol and sex hormone-binding globulin measures (SHBG), but with inconsistent direction of effects with regard to the NAFL risk allele (Fig. 3). The missense variants p.Thr165Ala in *MTARC1* and p.Ile43Val in *GPAM* had the greatest similarity in their associations with the tested traits (Figs. 3 and 4). The missense variant p.Cys325Gly in *APOH* was also associated with increased risk of atrial fibrillation ($P$ = 3.5 × 10$^{-9}$, OR = 1.12, $n_{cases}$ = 96,018), heart failure ($P$ = 2.4 × 10$^{-9}$, OR = 1.12, $n_{cases}$ = 99,214) and higher levels of lipoprotein (a) (Lp(a); $P$ = 2.2 × 10$^{-56}$, effect = 0.10 s.d.) (Supplementary Table 4). The p.His48Arg in *ADH1B* was associated with reduced risk

of alcohol dependence ($P$ = 2.6 × 10$^{-64}$, OR = 0.31, $n_{cases}$ = 60,800). We compared variant associations with alcoholic liver disease (ALD) diagnoses and NAFL diagnoses (Supplementary Fig. 4). The p.His48Arg was the only variant with a significantly greater effect on the risk of ALD ($n_{cases}$ = 3,818) than of NAFL (OR = 0.33 and OR = 0.85, respectively, $P$ = 6.6 × 10$^{-10}$).

### Rare loss-of-function variants in *GPAM* and *MTARC1*

Among the 20 variants with GWS associations with NAFL and cirrhosis, 14 are common or low-frequency missense variants (MAF < 3%). Although these variants implicate probable relevant genes, it is not clear whether loss or gain of function of the encoded proteins reduces or increases disease risk. To investigate this, we looked for associations with rare predicted loss-of-function (pLOF) variants in candidate causal genes at these loci, using data from Iceland and UK. We tested 47 pLOF variants for associations with the same traits that we detected with the lead variant at the locus and found two rare pLOF variant associations in Iceland: p.Arg305Ter in *MTARC1* (MAF = 0.34%) and p.Thr189GlyfsTer5 in *GPAM* (MAF = 0.11%) (Supplementary Table 5). Both genes encode mitochondrial enzymes[46–48]. The rare pLOF variant p.Arg305Ter in *MTARC1* was associated with lower total cholesterol levels ($P$ = 3.2 × 10$^{-5}$, effect = −0.18 s.d.) (Fig. 4 and Supplementary Table 5), consistent with the protective allele of the common missense variant p.Thr165Ala in *MTARC1*. As p.Arg305Ter is predicted to reduce the function of *MTARC1*, this suggests that p.Thr165Ala also lowers cholesterol levels through reduced *MTARC1* function. Therefore, its association with a reduced risk of NAFL is most probably driven by

**Table 2 | GWASs with NAFL (combination of PDFF and NAFL ICD-10 code diagnosis)**

| rs no. | Chromosome | Position (hg38) | Effect allele | Other allele | MAF (%) | Closest gene | Coding change | Protein change | P value, PDFF | Effect (s.d.), PDFF | P value, NAFL | OR, NAFL |
|---|---|---|---|---|---|---|---|---|---|---|---|---|
| rs738409 | chr22 | 43928847 | G | C | 21.63 | *PNPLA3* | Missense | p.Ile148Met | $3.02×10^{-217}$ | 0.28 | $9.73×10^{-116}$ | 1.47 |
| rs58542926 | chr19 | 19268740 | T | C | 7.42 | *TM6SF2* | Missense | p.Glu167Lys | $1.67×10^{-126}$ | 0.34 | $8.24×10^{-37}$ | 1.39 |
| rs187429064 | chr19 | 19269704 | G | A | 1.28 | *TM6SF2* | Missense | p.Leu156Pro | $1.47×10^{-32}$ | 0.39 | $1.43×10^{-9}$ | 1.42 |
| rs429358 | chr19 | 44908684 | C | T | 15.41 | *APOE* | Missense | p.Cys130Arg | $6.63×10^{-24}$ | −0.11 | $1.50×10^{-23}$ | 0.81 |
| rs28601761 | chr8 | 125487789 | G | C | 41.27 | *TRIB1* | Intergenic | | $3.46×10^{-21}$ | −0.072 | $2.13×10^{-15}$ | 0.89 |
| rs1260326 | chr2 | 27508073 | T | C | 38.80 | *GCKR* | Missense | p.Leu446Pro | $2.74×10^{-18}$ | 0.067 | $9.58×10^{-18}$ | 1.14 |
| rs2792751 | chr10 | 112180571 | T | C | 26.91 | *GPAM* | Missense | p.Ile43Val | $5.24×10^{-15}$ | 0.066 | $8.86×10^{-8}$ | 1.09 |
| rs13389219 | chr2 | 164672366 | T | C | 39.49 | *COBLL1* | Intergenic | | $1.87×10^{-12}$ | −0.054 | $6.89×10^{-5}$ | 0.94 |
| rs140201358 | chr11 | 823586 | G | C | 1.32 | *PNPLA2* | Missense | p.Asn252Lys | $4.88×10^{-12}$ | 0.22 | $1.31×10^{-2}$ | 1.16 |
| rs641738 | chr19 | 54173068 | T | C | 43.77 | *TMC4* | Missense | p.Gly17Glu | $7.36×10^{-12}$ | 0.052 | $2.69×10^{-6}$ | 1.07 |
| rs2642438 | chr1 | 220796686 | A | G | 29.12 | *MTARC1* | Missense | p.Thr165Ala | $7.70×10^{-11}$ | −0.053 | $6.53×10^{-13}$ | 0.89 |
| rs7029757 | chr9 | 129804387 | A | G | 9.23 | *TOR1B* | Intron | c.465+49G>A | $9.22×10^{-10}$ | −0.078 | $2.80×10^{-3}$ | 0.92 |
| rs1801689 | chr17 | 66214462 | C | A | 2.96 | *APOH* | Missense | p.Cys325Gly | $3.98×10^{-9}$ | 0.13 | $2.17×10^{-2}$ | 1.11 |
| rs1229984 | chr4 | 99318162 | T | C | 3.13 | *ADH1B* | Missense | p.His48Arg | $4.40×10^{-9}$ | −0.14 | $3.3×10^{-3}$ | 0.85 |
| rs11944752 | chr4 | 99564098 | A | G | 25.67 | *MTTP* | Upstream | | $4.22×10^{-8}$ | −0.047 | $5.54×10^{-7}$ | 0.92 |
| rs6955582 | chr7 | 65966699 | A | G | 44.92 | *GUSB* | Intron[a] | c.1653+1032T>C | $4.68×10^{-7}$ | −0.038 | $1.90×10^{-4}$ | 0.95 |
| rs1800562 | chr6 | 26092913 | A | G | 7.336 | *HFE* | Missense | p.Cys282Tyr | $5.90×10^{-7}$ | 0.071 | $1.00×10^{-3}$ | 1.10 |
| rs2862954 | chr10 | 100152307 | C | T | 47.86 | *ERLIN1* | Missense | p.Ile291Val | $3.06×10^{-5}$ | −0.031 | $3.15×10^{-6}$ | 0.93 |

[a]Correlated with a missense variant, p.Leu649Pro ($r^2$ = 0.87 in the UKB and 0.99 in Iceland). Associations are shown for the nonoverlapping sets of GWASs on PDFF and NAFL separately. Effects of variants on PDFF are shown as s.d. and on NAFL as ORs.

reduced *MTARC1* function. Our findings align with reports of another protein-truncating variant in *MTARC1*, p.Arg200Ter, that is associated nominally with reduced cirrhosis risk[20]. The minor allele of the common missense variant in *GPAM*, p.Ile43Val, is associated with increased risk of NAFL and higher total cholesterol levels, whereas the rare pLOF variant, p.Thr189GlyfsTer5, is associated with lower total cholesterol levels ($P = 3.5 × 10^{-8}$, effect = −0.41 s.d.), suggesting that p.Thr189GlyfsTer5 decreases NAFL risk through decreased *GPAM* function. Although neither of these pLOFs is associated with NAFL or cirrhosis, they give information on whether previously reported associations are gain or loss of function. *MTARC1*'s p.Arg305Ter is located in the gene's last exon, and we found no evidence of nonsense-mediated decay in our RNA data (Supplementary Fig. 3). We also found a pLOF variant in *GCKR*, p.Arg540Ter (MAF = 0.60%), which is associated with increased triglyceride levels ($P = 6.9 × 10^{-7}$, effect = 0.16 s.d.), similar to the NAFL risk-increasing allele of the *GCKR* common missense variant p.Leu446Pro.

**Variant–variant and variant–BMI interactions**

rs72613567[TA] in *HSD17B13* and p.Ile148Met in *PNPLA3* interact in their effects on liver enzymes[21]. We replicated the interaction between rs72613567[TA] in *HSD17B13* and p.Ile148Met in *PNPLA3* ($P = 9.0 × 10^{-15}$ and $2.9 × 10^{-12}$ on aspartate transaminase (AST) and ALT, respectively) (Supplementary Fig. 5). The interaction of these two variants on the diagnoses of NAFL and HCC was also significant ($P = 0.00073$ and $P = 0.0020$, respectively).

PDFF and body mass index (BMI) were correlated ($r^2 = 0.32$), with only 2% of individuals having high PDFF (>5%) and low BMI (<25 kg m$^{-2}$) compared with 19% with high PDFF and high BMI (≥25 kg m$^{-2}$) (Table 1 and Fig. 1). Among the 20 NAFL and cirrhosis variants, only p.Cys130Arg in *APOE* and p.His48Arg in *ADH1B* were associated with BMI (Supplementary Table 4, $n_{BMI measures} = 486,305$). Missense variants in *PNPLA3*, *TM6SF2*, *APOE* and *GUSB* interacted with BMI to affect PDFF ($P < 0.05/20$; Supplementary Table 6 and Supplementary Fig. 6). None of the variants interacted with age to affect PDFF (Supplementary Fig. 7).

Among participants in the UKB, 2,795 had 2 liver MR images taken 2–3 years apart. Consecutive PDFF measurement were correlated ($r^2 = 0.90$), as were BMI measurements ($r^2 = 0.60$) (Supplementary Figs. 8 and 9). Changes in PDFF and BMI associated strongly ($P = 2.8 × 10^{-104}$), with direction of change in the two measures that on average were the same (Supplementary Fig. 10). We found a nominally significant interaction between p.Ile148Met in *PNPLA3* and changes in BMI ($P = 0.030$; Supplementary Fig. 11). Furthermore, the variations in the PDFF changes were greater among p.Ile148Met carriers than noncarriers ($P = 0.0055$; Supplementary Fig. 12) and were not fully explained by the interaction between the genotype and change in BMI ($P$ adjusted for change in BMI: $P_{adj} = 0.032$).

**Integration with circulating protein levels**

To gain insight into proteins affecting the pathogenesis of NAFL or cirrhosis and search for relevant biomarkers, we analyzed protein levels measured with 4,907 aptamers by SomaScan[49] v.4 in 35,559 Icelanders[50] and 1,459 immunoassays using the Olink Explore 1536 in 47,151 European-ancestry participants from the UKB. The levels of 2,741 proteins associated (after Bonferroni's adjustment) with NAFL ($n_{Iceland} = 234$ cases, $n_{UK} = 572$ cases) and 948 with cirrhosis ($n_{Iceland} = 111$ cases, $n_{UK} = 303$ cases) in either Iceland or UKB.

We looked for associations between the 20 NAFL and cirrhosis variants and protein levels. Sixteen variants were associated with the level of one or more proteins in *trans*, using either Iceland (SomaScan) or UKB (Olink), as top protein QTLs (pQTLs) (Supplementary Table 7). The *trans* pQTLs in *GCKR* and *SERPINA1* are nonspecific in that they are associated with 396 and 172 proteins, respectively. Focusing on the variants associated with fewer than 100 proteins, we identified 273 proteins that associated with the variants in *trans* (that is, variants at

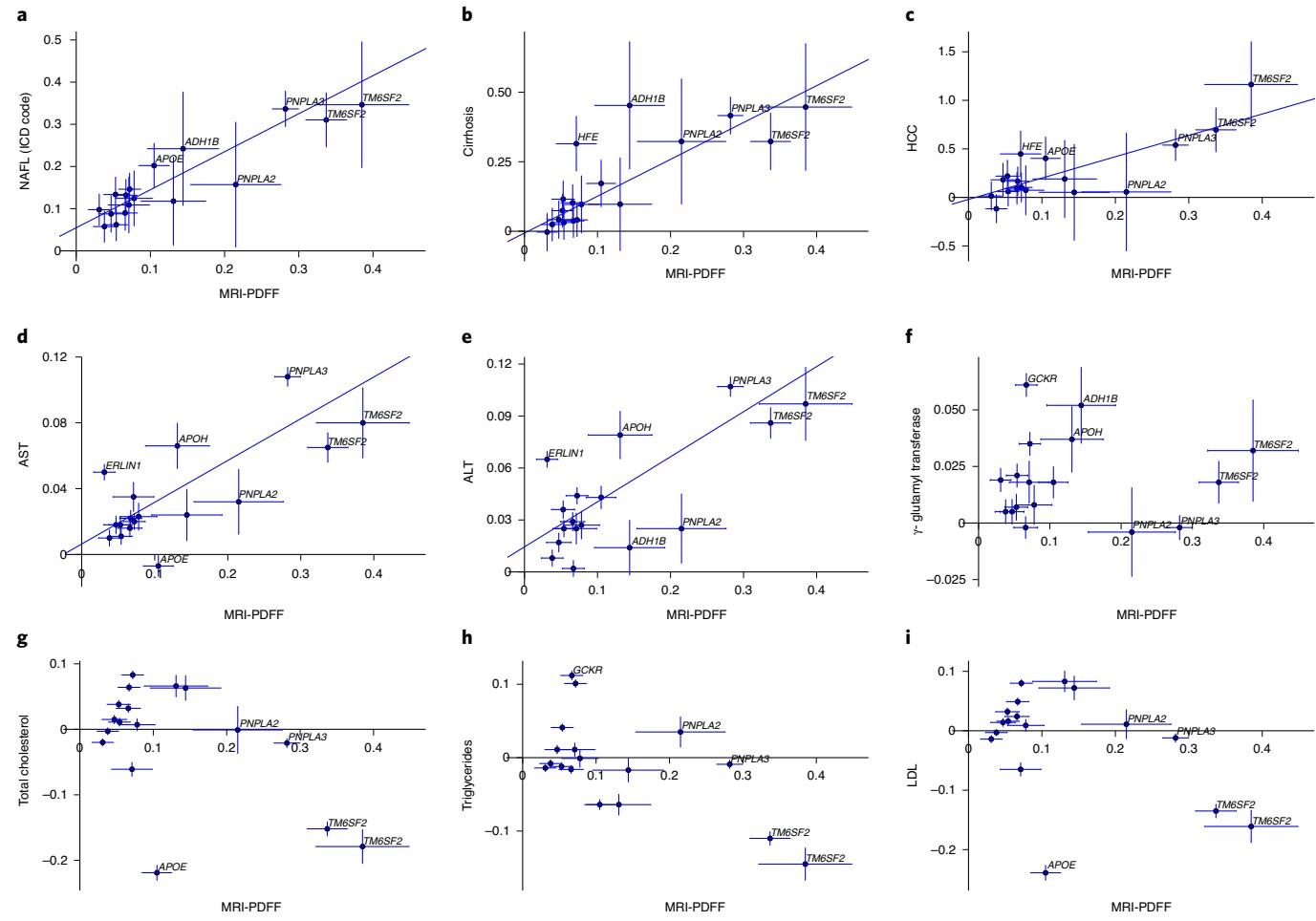

**Fig. 2 | Effects of sequence variants on PDFF compared with effects on liver disease, liver enzymes and lipids. (a–i)** Effects of sequence variants on PDFF ($n = 36,116$) compared with their effects on ICD-10 code-diagnosed NAFL ($n_{cases} = 5,921$) (**a**), cirrhosis ($n_{cases} = 2,301$) (**b**), HCC ($n_{cases} = 374$) (**c**) and measures of AST ($n = 389,272$) (**d**), ALT ($n = 390,519$) (**e**), γ-glutamyl transferase ($n = 390,457$) (**f**), total cholesterol ($n = 390,652$) (**g**), triglycerides ($n = 390,346$) (**h**) and low-density lipoprotein ($n = 389,974$) (**i**) in the UKB. The effects (box of error bars) and their 95% confidence intervals (CIs) (error bars) are shown for the allele that increases PDFF and either in s.d. for quantitative traits or as log(OR) for binary traits.

**Table 3 | GWASs with cirrhosis**

| P value combined | OR | P heterogeneity | rs no. | Chromosome | Position (hg38) | Effect allele | Other allele | Closest gene | Coding change | Protein change | P values per cohort | Effects per cohort | MAF (%) per cohort |
|---|---|---|---|---|---|---|---|---|---|---|---|---|---|
| $1.63×10^{-84}$ | 1.60 | 0.00080 | rs738409 | chr22 | 43928847 | C | G | *PNPLA3* | Missense | p.Ile148Met | $1.92×10^{-28}$, $3.70×10^{-35}$, $1.01×10^{-8}$, $6.51×10^{-21}$ | 2.03, 1.50, 1.60, 1.56 | 23.00, 21.63, 22.09, 22.73 |
| $1.24×10^{-14}$ | 1.70 | 0.83 | rs28929474 | chr14 | 94378610 | T | C | *SERPINA1* | Missense | p.Glu366Lys | 0.022, $5.66×10^{-8}$, 0.00074, 0.00016 | 1.87, 1.64, 1.99, 1.70 | 0.81, 1.89, 2.17, 2.00 |
| $2.00×10^{-14}$ | 1.33 | 0.25 | rs58542926 | chr19 | 19268740 | T | C | *TM6SF2* | Missense | p.Glu167Lys | $6.71×10^{-5}$, $4.15×10^{-10}$, 0.12, 0.022 | 1.50, 1.38, 1.22, 1.20 | 7.69, 7.42, 7.80, 6.45 |
| $6.71×10^{-9}$ | 0.87 | 0.34 | rs72613567 | chr4 | 87310240 | TA | T | *HSD17B13* | Splice region | c.704+2dup | 0.162, 0.00058, 0.11, $5.81×10^{-6}$ | 0.91, 0.89, 0.88, 0.81 | 29.35, 26.96, 27.78, 21.87 |

Associations are shown for all four studies (deCODE genetics, UKB, Intermountain and FinnGen).

a different chromosomal location to the gene encoding the targeted protein), of which 26 were associated with both SomaScan and Olink measurements. Levels of ten proteins were associated with five or more variants (SMPDL3A and NAAA proteins with eight variants; SMPD1 and GUSB protein with seven variants; KRT18, HEXB, GSTA1, ENTPD5, CTSO and ACY1 proteins with five variants) (Supplementary Fig. 15) and eight out of these ten proteins are associated with NAFL or cirrhosis (Supplementary Tables 8–10).

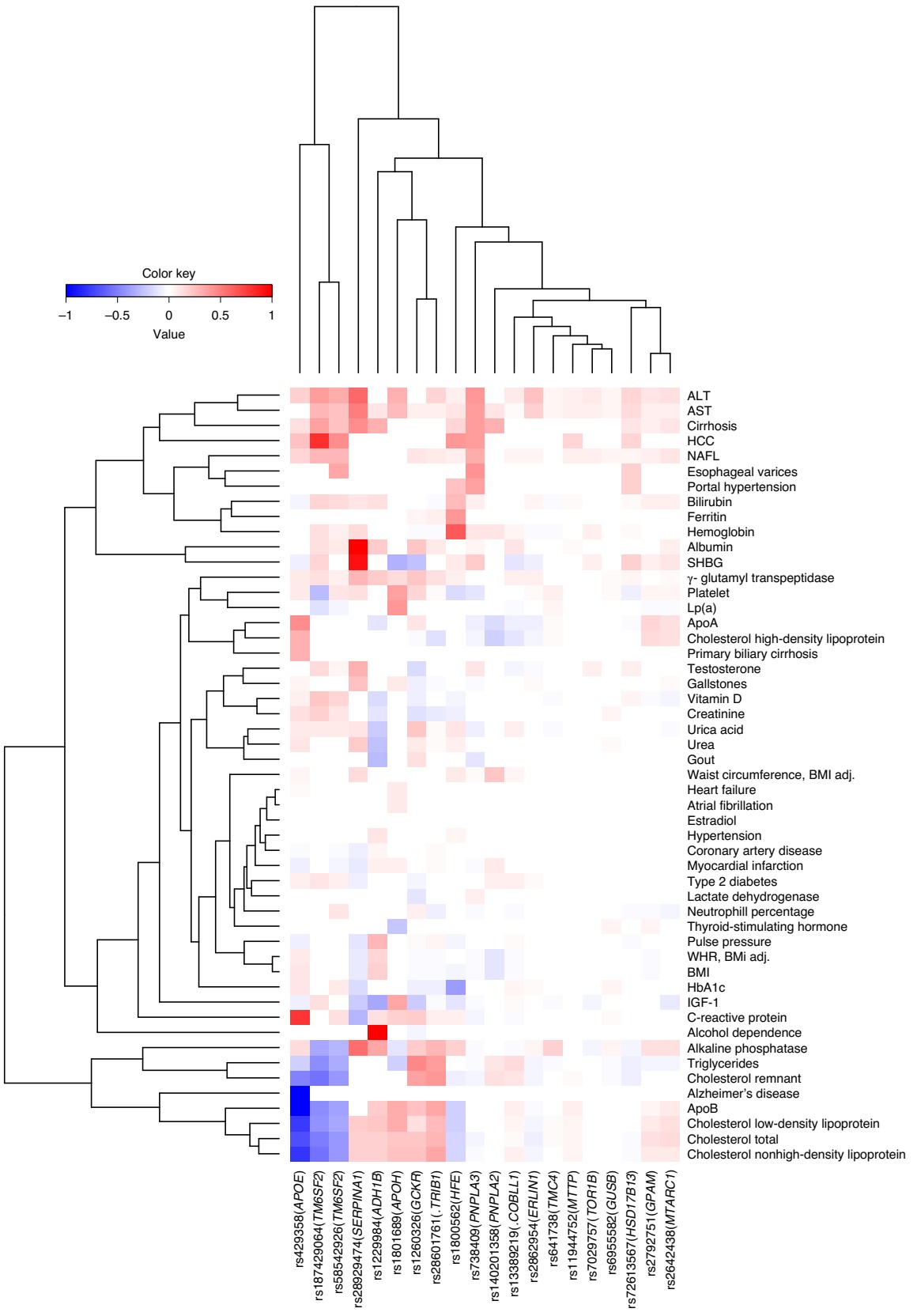

**Fig. 3 | Pleiotropic effects of NAFLD variants.** The effects of the identified variants with 51 phenotypes, including liver enzymes and lipid levels. The effect (colored from red to blue) on each phenotype is shown for the allele that increases PDFF and risk of NAFL or cirrhosis and is scaled to the range of [−1:1] for binary and quantitative phenotypes separately. Effects are shown only for the significant associations after an FDR correction. Apo, Apolipoprotein; HbA1c, glycated hemoglobin.

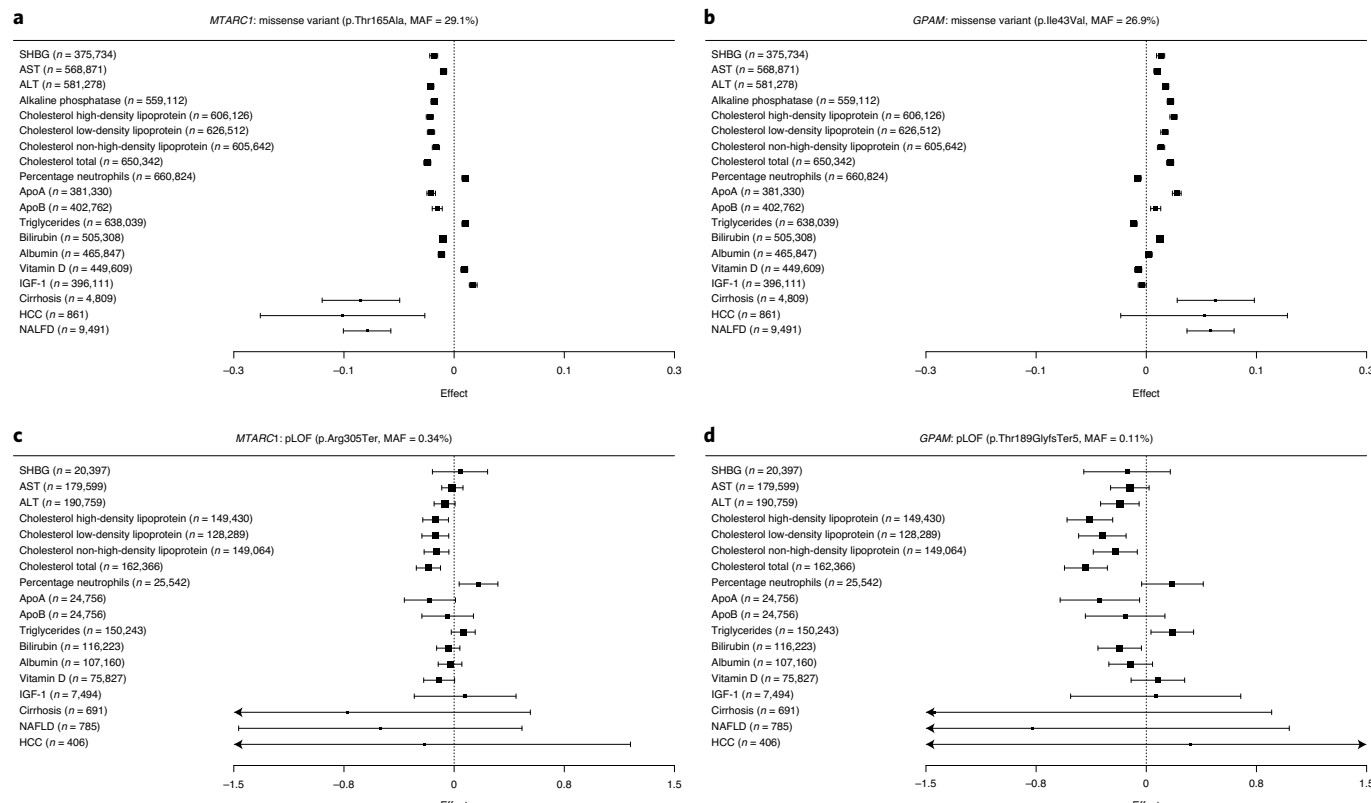

**Fig. 4 | Association pattern of missense and loss-of-function variants in *GPAM* and *MTARC1*. (a,b)** Plots show a similar association pattern of common missense variants in *MTARC1* (**a**) and *GPAM* (**b**) with selected phenotypes. On the *x* axes, the effects of the minor alleles (box of the error bars) and their corresponding 95% CIs (error bars) are shown (log(OR) for binary traits). (**c,d**) Plots show the association pattern of rare predicted loss-of-function variants identified in the Icelandic population with the corresponding phenotypes in Iceland for *MTARC1* (**c**) and *GPAM* (**d**). The OR of p.Thr189GlyfsTer5 for cirrhosis is 0, which is outside the plotting region. The *n* measures are shown for quantitative traits and *n* cases for binary traits. The data used to generate these plots are presented in Supplementary Tables 4 and 5.

The missense variant p.Ile148Met in *PNPLA3* is a top *trans* pQTL for 86 proteins, with aldo–keto reductase family 7 member 3 protein (AKR7A3) showing the most significant association in Iceland (SomaScan) ($P = 3.1 \times 10^{-12}$, effect = 0.07 s.d.) and keratin 18 (KRT18) in UKB (Olink) ($P = 8.4 \times 10^{-38}$, effect = 0.10 s.d.). The rs72613567[TA] in *HSD17B13* was associated most significantly with levels of the DnaJ B member homolog subfamily 9 protein (DNAJB9) in Iceland (SomaScan) and carboxylesterase 3 (CES3) in UKB (Olink). Missense variants in *MTARC1* (p.Thr165Ala) and *GPAM* (p.Ile43Val) were both associated with levels of group XIIB secretory phospholipase A2-like protein (PLA2GXIIB) as measured in Iceland (SomaScan), and inactive Cα-formylglycine-generating enzyme (SUMF2), hypoxia-upregulated protein 1 (HYOU1) and acid sphingomyelinase-like phosphodiesterase 3a (SMPDL3A), as measured in the UKB (Olink). PLA2GXIIB is highly expressed in the liver and the NAFL protective allele of these variants was associated with lower protein levels ($P = 9.3 \times 10^{-12}$, effect = −0.062 s.d. and $P = 2.4 \times 10^{-14}$, effect = −0.072 s.d., for *MTARC1* and *GPAM* variants, respectively). Moreover, the rare pLOF variants in Iceland in *GPAM* and *MTARC1* were both associated with lower PLA2GXIIB protein levels ($P = 7.4 \times 10^{-4}$, effect = −0.44 s.d. and $P = 0.013$, effect = −0.19 s.d., respectively). We note that other NAFL variants were associated nominally with PLA2GXIIB, some with an opposite direction of effects to the LOF variants in *GPAM* and *MTARC1* (Supplementary Fig. 13). Therefore, to explore whether any protein measured in plasma may have a causal role in disease, we performed Mendelian randomization using all-*trans* pQTLs for each protein. The effects of pQTLs of the transferrin receptor protein 1 (TFRC) were proportional to their effect on PDFF ($P_{ivw-olink} = 5.6 \times 10^{-10}$,

$P_{ivw-somascan} = 2.0 \times 10^{-4}$; Supplementary Fig. 13 and Supplementary Table 11), suggesting that TFRC may have a causal role in NAFL. Apart from TFRC, the analysis suggests that the alterations in protein levels in plasma are not causal but rather a consequence of disease because, for many proteins, the effects of the set of NAFL variants on PDFF were proportional to their effects on protein level (Supplementary Table 12).

We performed a pathway analysis using the PANTHER v.16.0 tool to seek understanding of the variant–liver disease associations. The 273 proteins associating with NAFL variants were enriched for multiple metabolic and catabolic processes, including the metabolism of hormones, lipids, alcohol, vitamins, steroids and monocarboxylic acid among other pathways and biological processes (Supplementary Table 13).

Last, we compared the plasma proteome of those diagnosed with cirrhosis with those with NAFL (Supplementary Table 10 and Supplementary Fig. 14). In Iceland (SomaScan), the most significant difference between NAFL and cirrhosis was, first, for calsyntenin 2 (CSTN2, $P = 3.6 \times 10^{-17}$, effect = 0.92 s.d.) and, second, for insulin-like growth factor-binding protein (IGFBP) 2 ($P = 5.9 \times 10^{-17}$, effect = 0.92 s.d.). IGFBP2 levels were elevated in individuals with cirrhosis but reduced in NAFL compared with population controls. Using UKB (Olink), the most significant difference between NAFL and cirrhosis was, first, for thrombospondin 2 (THBS2, $P = 6.6 \times 10^{-114}$, effect = 1.59 s.d.) and, second, for IGFBP7 ($P = 1.3 \times 10^{-89}$, effect = 1.35 s.d.). Levels of THBS2, IGFBP7 and IGFPB2 are measured with both SomaScan and Olink and are associated in both datasets. For NAFL compared with the population,

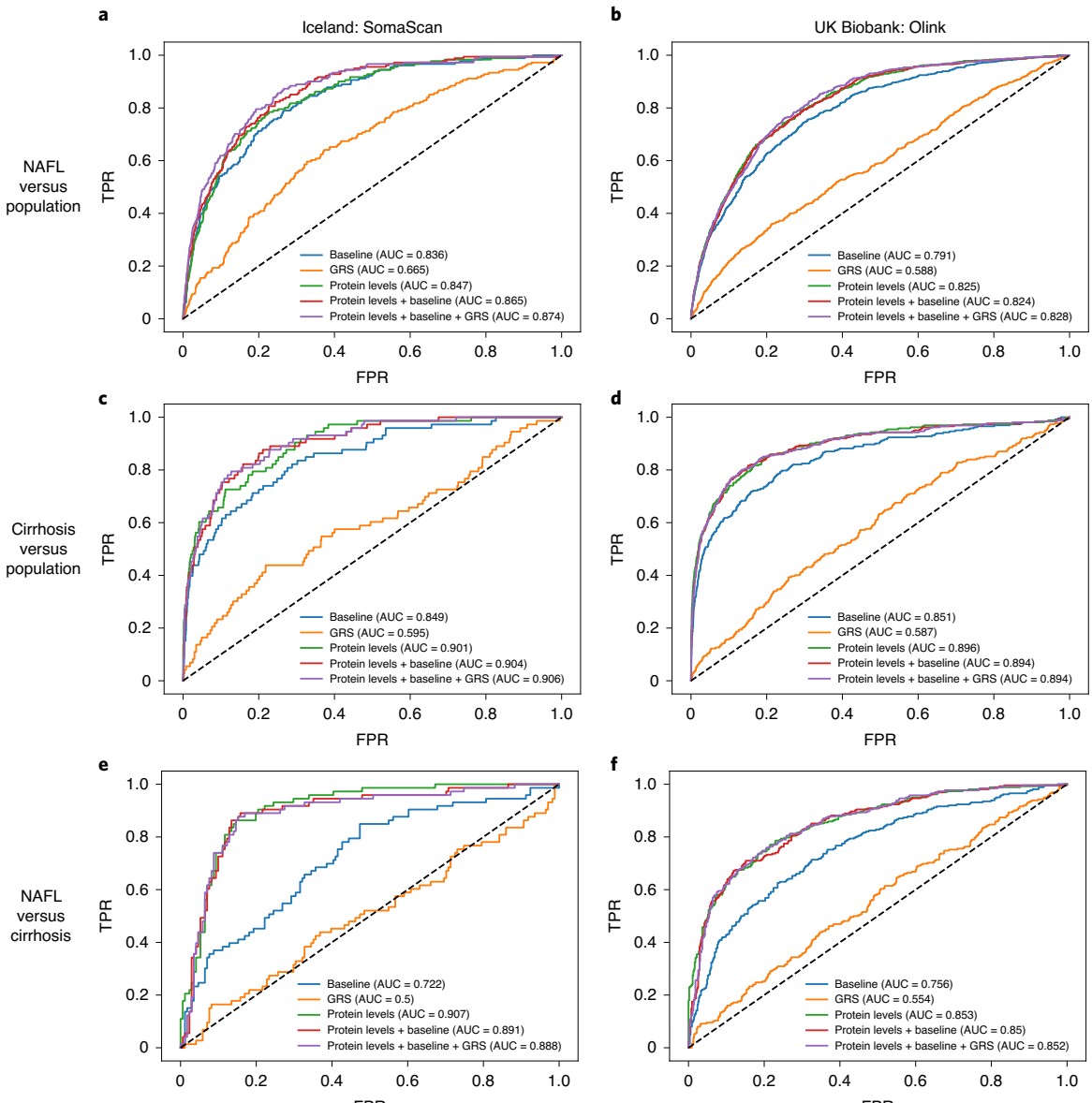

**Fig. 5 | ROC for models trained to discriminate between NAFL and cirrhosis.** **a–f**, ROC AUCs for models trained to discriminate between NAFL and population (**a,b**), cirrhosis and population (**c,d**), NAFL and cirrhosis (**e,f**). **a,c,e** Plots show results for the SomaScan data using the Icelandic population. **b,d,f** Plots show results for the Olink data using the UKB population. The SomaScan analysis was performed on 181 individuals with NAFL and 73 with cirrhosis, and the Olink

analysis was performed on 610 individuals with NAFLD and 262 with cirrhosis. Models trained to discriminate between the presence of disease diagnosis and population were trained, respectively, on an additional set of 20,619 individuals (SomaScan) and 38,018 individuals (Olink) without cirrhosis and NAFL. FPR, false-positive rate; TPR, true-positive rate.

aminoacylase-1 (ACY1) showed the strongest association in both Iceland and UKB (Supplementary Tables 8–10).

To investigate whether plasma proteins can effectively discriminate between having an NAFL and cirrhosis, we trained penalized logistic regression models using liver enzymes, age, sex and BMI as a baseline, as well as using the plasma proteome and genetic risk scores (GRSs). In both Iceland (SomaScan) and UKB (Olink), the models trained with the plasma proteome outperformed other models in discriminating between NAFL and cirrhosis, NAFL and the population, and cirrhosis and the population (Fig. 5 and Supplementary Tables 14–17).

## Discussion
We identified 18 sequence variants that are associated with NAFL and two additional variants that are associated with cirrhosis and not PDFF.

The NAFL variants affect the risk of cirrhosis and HCC proportional to their effects on PDFF, supporting a causal relationship between hepatic fat accumulation and these diseases[51,52].

The NAFLD variants are associated with other phenotypes, with variable patterns of association. One interpretation of this is that perturbations of more than one biochemical pathway lead to NAFLD. The strongest NAFL-associated variant, p.Ile148Met, in *PNPLA3* interacts with rs72613567[TA] in *HSD17B13* and BMI but not with age. Longitudinal PDFF measures suggest that p.Ile148Met carriers are more susceptible to change in BMI than noncarriers, in line with previous studies suggesting that the p.Ile148Met genotype increases sensitivity to the beneficial effects of dietary interventions and bariatric surgery[53–56].

Sixteen NAFL-associated variants were coding, five were eQTLs and three sQTLs, and many implicated genes were involved in lipid metabolism, reinforcing the notion of its importance as a primary process in

NAFL pathogenesis[57,58]. For example, variants at *TM6SF2* exert their effects on hepatic lipid concentration by reducing *TM6SF2* function, causing reduced secretion of triglyceride-rich lipoproteins[59]. Several variants affect NAFL and blood lipids in the same direction, including p.Leu446Pro in *GCKR*, which also decreases glycated hemoglobin and type 2 diabetes risk. This is consistent with evidence that the variant increases hepatic glucose metabolism and de novo lipogenesis[60]. The p.His48Arg in *ADH1B* is associated with less risk of having an NAFL diagnosis. The variant associates with less alcohol consumption and reduces the risk of ALD substantially more than the risk of NAFL. The amount of consumed alcohol (2 and 3 units of alcohol for women and men, respectively) used to distinguish between NAFL and ALD is quite arbitrary[61]. Therefore, it is likely that the association of the variant with NAFL diagnosis is driven by its effect on alcohol consumption below the ALD diagnosis threshold. None of the other NAFL variants was associated with a significantly greater effect on ALD than NAFL.

The NAFL associations include missense variants in *GPAM* and *MTARC1*, which both encode mitochondrial enzymes and are highly expressed in liver and adipose tissue[31]. *GPAM* encodes glycerol-3-phosphate acyltransferase 1, which catalyzes the first step of triglyceride synthesis. *Gpam* knockout mice have lower hepatic triglyceride content and overexpression has the opposite effect[48,62]. The results for *MTARC1* are in line with previous reports of the effects of missense and LOF variants in *MTARC1*, but the mechanism explaining the associations was less clear[20]. The similarity in their associations with other traits indicates that mutual or similar pathways explain the *MTARC1* and *GPAM* associations. Furthermore, both missense variants affect plasma levels of PLA2GXIIB protein. We identified rare pLOF variants in both genes affecting the same traits and reducing liver enzymes and total cholesterol levels. This suggests that *GPAM* and *MTARC1* have an etiological role and inhibition of *GPAM* and *MTARC1* could be therapeutic for NAFL or NASH, with a favorable effect on the metabolic profile. The lack of associations with increased risk of a large set of diseases is reassuring with regard to treatment safety.

Among associations with NAFL are missense variants in *APOH* and *GUSB*. *APOH* is highly and almost exclusively expressed in the liver[31]. The other trait associations of p.Cys325Gly in *APOH* strongly suggest a role in lipid metabolism[35,63,64]. Furthermore, the variant has been reported in GWASs of coronary artery disease[65] and Lp(a) levels[64]. We replicated the Lp(a) association and observed associations with an increased risk of atrial fibrillation and heart failure. Our results indicate a role for *GUSB* in the etiology of NAFL. The missense variant p.Leu649Pro in *GUSB* associates with both NAFL and RNA expression levels of *GUSB*. Furthermore, seven NAFL variants associate with GUSB plasma protein levels. *GUSB* encodes β-glucuronidase, a lysosomal enzyme involved in the breakdown of glycosaminoglycans[66].

Plasma proteome analysis revealed that missense and pLOF variants in *MTARC1* and *GPAM* are associated with levels of PLA2GXIIB in plasma. PLA2GXIIB is highly expressed in the liver and knockout mice have severe hepatosteatosis[67]. However, our proteomic analysis does not support an etiological role for PLA2GXIIB because NAFL variants affect plasma PLA2GXIIB levels with effect directions inconsistent with their NAFL effects. Evidence suggests that PLA2GXIIB is a mediator of hepatic lipoprotein secretion and its inhibition results in decreased levels of plasma lipids[67,68]. Thus, PLA2GXIIB may mediate variant effects on circulating lipids without directly affecting hepatic fat and thus could serve as a biomarker of drug target engagement.

Diagnosis and monitoring of complications in patients with NAFLD are challenging[2]. We designed models including plasma proteins that outperformed a model trained on liver enzymes and GRSs in discriminating between NAFL and cirrhosis diagnoses. Thus, levels of plasma proteins have the potential to serve as a noninvasive tool for use in the diagnosis and monitoring of disease, whereas GRSs are associated with a lifetime risk of disease. THBS2 was elevated in individuals with cirrhosis compared with NAFL and the population, and ACY1 in

individuals with NAFL compared with the general population. Intrahepatic *THBS2* expression levels have previously been shown to correlate with fibrosis in patients with NAFLD[69]. The association of IGFBP2 and IGFBP7 with cirrhosis and NAFL is consistent with previous studies of NASH progression[70]. Both proteins bind insulin-like growth factors (IGFs) and modulate their availability[71,72]. IGFs are mainly produced in the liver, and IGFBP2 and IGFBP7 elevation may reflect imbalances in the IGF system due to liver damage[73]. However, an etiological role has been suggested for IGFBP7, which may contribute to hepatic fibrogenesis[74] and act as a tumor suppressor in HCC[75]. Levels of SHBG are also associated with cirrhosis compared with NAFL, consistent with previous reports of a positive correlation with advanced fibrosis in NASH[70]. There are conflicting epidemiological studies about whether NAFLD is associated with increased or decreased levels of SHBG. In line with this, many NAFL variants are associated with SHBG plasma levels with inconsistent directions of effect compared with their effect on hepatic fat content[76].

A limitation of the present study is the lack of data enabling a more detailed phenotype stratification, in particular histological data for disease staging. Furthermore, information on other causes of liver disease, such as alcohol consumption, is limited. Our approach is, however, in line with the recent opinion to not base the disease diagnosis on a criterion of exclusion of other diseases, such as ALD[77].

In conclusion, we used multiomics data to shed light on the genetic basis of NAFLD. Analysis of pLOF variants, blood RNA expression and plasma proteomics data pointed to causative genes and whether changes in their functions contribute to the pathogenesis. We demonstrated the diverse effects of NAFL risk alleles on other diseases and traits, including blood lipids and proteins, and showed that plasma proteomics has the potential to stage NAFLD.

## Online content

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

[1]deCODE genetics/Amgen, Inc., Reykjavik, Iceland. [2]Faculty of Electrical and Computer Engineering, University of Iceland, Reykjavik, Iceland. [3]Faculty of Medicine, University of Iceland, Reykjavik, Iceland. [4]Internal Medicine and Emergency Services, Landspitali—The National University Hospital of Iceland, Reykjavik, Iceland. [5]Röntgen Domus, Reykjavík, Iceland. [6]Children's Medical Center, Landspítali—The National University Hospital of Iceland, Reykjavík, Iceland. [7]Department of Clinical Immunology, Aarhus University Hospital, Aarhus, Denmark. [8]Intermountain Healthcare, St. George, UT, USA. [9]Intermountain Healthcare, Salt Lake City, UT, USA. [10]Faculty of Mechanical Engineering, Industrial Engineering and Computer Science, University of Iceland, Reykjavik, Iceland. [11]Clinical Laboratory Services, Diagnostics and Blood Bank, Landspitali—The National University Hospital of Iceland, Reykjavik, Iceland. [12]Department of Clinical Immunology, Copenhagen University Hospital, Rigshospitalet, Cophenhagen, Denmark. [13]Department of Clinical Medicine, University of Copenhagen, Copenhagen, Denmark. [14]Department of Clinical Immunology, Zealand University Hospital, Køge, Denmark. [15]Development Centre for Primary Health Care in Iceland, Reykjavík, Iceland. [16]Department of Family Medicine, University of Iceland, Reykjavík, Iceland. [17]Department of Cardiology, The Heart Centre, Copenhagen University Hospital, Rigshospitalet, Copenhagen, Denmark. [18]Amgen, Cardiometabolic Disorders, South San Francisco, CA, USA. [19]School of Engineering and Natural Sciences, University of Iceland, Reykjavik, Iceland. [20]These authors contributed equally: Gardar Sveinbjornsson, Magnus O. Ulfarsson. ✉e-mail: gardars@decode.is; kstefans@decode.is

**DBDS Genomic consortium**

**Christian Erikstrup**[7], **Ole Birger Pedersen**[13,14], **Unnur Thorsteinsdottir**[1,3], **Daniel F. Gudbjartsson**[1,2,19], **Hilma Holm**[1] and **Kari Stefansson**[1,3]

A list of members and their affiliations appears in the Supplementary information.

## Methods

### Study populations

The UKB project is a large prospective cohort study of ~500,000 individuals from across the United Kingdom, aged between 40 and 69 years at recruitment[78]. The present study has collected extensive phenotypic and genotypic information on participants, including ICD-10-coded diagnoses from inpatient and outpatient hospital episodes and abdominal MRI through its imaging study[79]. Genotype imputation data were available for 487,409 participants (May 2017 release), of which 408,658 were included because they self-reported as white[80]. The UKB resource was used under application no. 56270. All phenotype and genotype data were collected following an informed consent obtained from all participants. The North West Research Ethics Committee reviewed and approved UKB's scientific protocol and operational procedures (REC reference no.: 06/MRE08/65).

The Icelandic deCODE genetics study is based on WGS data from 49,708 Icelanders participating in various research projects at deCODE genetics. Variants identified through WGS were imputed into 155,250 chip-genotyped Icelanders using long-range phasing[81] and their untyped close relatives based on genealogy. Sequencing was done using GAIIx, HiSeq, HiSeqX and NovaSeq Illumina technology. SNPs and insertions/deletions (indels) were identified, and their genotypes were called using joint calling with Graphtyper[82–84]. All participants who donated blood signed an informed consent. The personal identities of the participants and biological samples were encrypted by a third-party system. The study was approved by the Icelandic Data Protection Authority and the National Bioethics Committee of Iceland (no. VSN-20-182).

FinnGen summary statistics, including fatty liver disease and cirrhosis, were imported in December 2020 from a source available to researchers (v.4: https://www.finngen.fi/en/access_results) and methods have been documented (https://finngen.gitbook.io/documentation). The FinnGen database consists of samples collected from the Finnish biobanks and phenotype data collected at the national health registers. The Coordinating Ethics Committee of the Helsinki and Uusimaa Hospital District evaluated and approved the FinnGen research project. The project complies with existing legislation (in particular, the Biobank Law and the Personal Data Act). The official data controller of the present study is the University of Helsinki.

The Copenhagen Hospital Biobank Cardiovascular Study (CHB-CVDC) was used to acquire secondary cardiovascular phenotypes. CHB-CVDC involves a targeted selection of patients with cardiovascular disease from the CHB, a biobank based on patient blood samples drawn in Danish hospitals[43]. The CHB-CVDC has been approved by the National Committee on Health Research Ethics (1708829) and the Danish Data Protection Agency (P-2019-93). For binary phenotypes, the control group included blood donors from the Danish Blood Donor Study (DBDS) (n = 99,000), approved by the Danish Data Protection Agency (P-2019-99) and the Scientific Ethical Committee system (NVC 1700407)[44]. Chip typing and genotype imputation of CHB-CVDC and DBDS were performed at deCODE genetics using a north European sequencing panel of 15,576 individuals (including 8,429 Danes).

The samples from the USA (Intermountain dataset) were whole-genome studied using NovaSeq Illumina technology (n = 8,288) and genotyped using Illumina GSA chips (n = 28,279). Samples were filtered on a 98% variant yield. Over 245 million high-quality sequence variants and indels, to a mean depth of at least 20×, were identified using Graphtyper[82]. Quality-controlled chip genotype data were phased using Shapeit 4 (ref. [85]). A phased haplotype reference panel was prepared from the sequence variants using the long-range phased chip genotype data. The haplotype reference panel variants were then imputed into the chip-genotyped samples using inhouse tools and methods described previously[83,84]. In the US association analysis, samples assigned <93% CEU ancestry (northern European from Utah) were excluded. We adjusted for sex, age and the first 20 principal components. Phenotypic data were based on ICD-10 code diagnoses of individuals. The Intermountain Healthcare Institutional Review Board approved the study and all participants provided written informed consent before enrollment.

### Imaging protocol

The MR images used for calculating PDFF for 36,116 individuals were collected as a part of the UKB abdominal protocol, which, in turn, is part of the UKB imaging enhancement[79]. Two acquisitions were used, a single-slice GRE sequence and a single-slice IDEAL sequence[86]. The slice was captured through the center of the liver superior to the porta hepatis. The GRE sequence was captured using the following settings: repetition time (TR) = 27 ms, time to echo (TE) = [2.38, 4.76, 7.15, 9.53, 11.91, 14.29, 16.67, 19.06, 21.44, 23.82] ms, bandwidth = 710 Hz, flip angle (FA) = 20%, voxel size = $2.5 \times 2.5 \times 6.9$ mm$^3$ and a $160 \times 160$ acquisition matrix. The IDEAL sequence used TR = 14 ms, TE = [1.2, 3.2, 5.2, 7.2, 9.2, 11.2] ms, bandwidth = 1,565 Hz, FA = 5%, voxel size = $1.719 \times 1.719 \times 10.0$ mm$^3$ and a $256 \times 232$ acquisition matrix.

### Imaging analysis

We used two different approaches for calculating the PDFF from the liver MR images depending on whether the acquisition was GRE (n = 8,448) or IDEAL (n = 27,668). We implemented the PDFF estimation methods using a tailored Python code. For the GRE acquisition, we used a three-point Dixon method[87] to compute a PDFF map using the second, fourth and sixth echoes. Eight 25-voxel rectangular regions of interest (ROIs) were defined within the liver and we computed the mean and s.d. of the PDFF maps over those ROIs. The reported PDFF was the ROI with the lowest s.d. By choosing the lowest s.d., we avoided ROIs with water–fat swaps. For the IDEAL acquisition, we assumed the following signal model[88] for each voxel:

$$y_i = \left(\rho_w + \rho_f e^{-j2\pi\Delta f t_i}\right) e^{-j2\pi\varphi t_i} e^{-R_2^* t_i} + \varepsilon_i, i = 1, \dots, 6$$

where $\rho_w$ and $\rho_f$ are the water and fat components, respectively, $\Delta f$ is the chemical shift of fat with respect to water, $\varphi$ quantifies $B_0$ field inhomogeneity, $R_2^*$ is an MRI relaxation constant and $t_i$ the $i$th echo time. The parameters of interest, $\rho_w$, $\rho_f$ and $R_2^*$, are estimated from the signal model using an iterative weighted least-squares algorithm. The PDFF map was finally constructed using $|\rho_f|/(|\rho_w| + |\rho_f|)$ at each voxel. The reported PDFF was calculated in the same way as for the GRE case. Iron concentration (Supplementary Fig. 1) can be estimated by using the $R_2^*$ coefficient[89]:

Iron concentration = 0.202 + 0.0254 $R_2^*$.

To evaluate the correspondence between the PDFF scores for the IDEAL and the GRE acquisition, we investigated 1,222 PDFF scores computed for both. We also fit a linear model to quantify this relationship yielding in units of percentage:

$$\text{PDFF}_{\text{IDEAL}} \approx 0.26 + 0.89\, \text{PDFF}_{\text{GRE}}$$

and $R^2 = 0.92$. There are 3,869 PDFF scores for the GRE acquisition that we computed and that are available in the UKB data showcase. Supplementary Fig. 2 shows a scatterplot of those two sets of scores demonstrating good agreement. The relationship is given in units of percentage:

$$\text{PDFF}_{\text{GRE-UKB}} = -0.5935 + 1.0152\, \text{PDFF}_{\text{GRE}}$$

and $R^2 = 0.96$.

### Liver disease phenotype description

The NAFL sample consisted of 9,491 individuals from deCODE genetics, UKB, Intermountain and FinnGen. Case status was based on the ICD-10 code K76.0 (nonalcoholic fatty liver disease) from electronic

health records. To analyze NAFL complications, we defined an all-cause cirrhosis phenotype based on cirrhosis- and fibrosis-related ICD-10 codes (K70.2, K70.3, K70.4, K74.0, K74.1, K74.2, K74.6, K76.6 and KI85) and used ICD-10 code C22.0 for HCC. Analyzing this cirrhosis/fibrosis phenotype has previously been shown to increase power to detect associations compared with subtypes of nonalcoholic and alcoholic cirrhosis[20].

### Secondary phenotypes

NAFL-associated variants were tested for association with other phenotypes from deCODE genetics, UKB, FinnGen, CHB-CVDC/DBDS or other publicly available data sources, which contain extensive medical information on various traits. A total of 51 phenotypes was chosen for the analysis (Supplementary Table 4) based on known associations with NAFLD, liver function or the identified variants. These included other liver diseases and known blood markers of liver function, blood lipids, cardiovascular diseases, diabetes, anthropometric traits, hematological traits and hormone levels.

### Association testing and meta-analysis

We used logistic regression to test for association between sequence variants and binary phenotypes assuming an additive genetic model. For deCODE genetics, the model included the following covariates: sex, county of birth, current age or age at death (first- and second-order terms included), blood sample availability for the individual and an indicator function for the overlap of the lifetime of the individual with the time span of the phenotype collection. In CHB-CVDC/DBDS, the covariates were sex, age and 20 principal components to adjust for population stratification and blood sample availability. In the UKB study, 40 principal components were used to adjust for population stratification, and age and sex were included as covariates in the logistic regression model. When analyzing PDFF, BMI was included as a covariate in the analysis to increase power of associations. We used a linear mixed model implemented in BOLT-LMM to test for association between sequence variants and quantitative traits. The measurements used were adjusted for sex, year of birth and age at measurement (when available), and these were subsequently standardized to have a normal distribution. For the meta-analysis of summary-level statistics from different populations, we used a fixed-effects inverse variance method based on effect estimates and s.e.m. With the aim of studying NAFL, we combined summary-level data from the GWAS of PDFF and the meta-analysis of GWAS using ICD-10-code-based NAFL with multitrait analysis of genome-wide association summary statistics (MTAG)[28]. To have two nonoverlapping sets, we excluded individuals from the NAFL ICD-10 code analysis who had a PDFF measurement in the UK. To account for inflation in test statistics due to cryptic relatedness and stratification, we applied the method of LD score regression[90]. For the GWS, we accounted for multiple testing with a weighted Bonferroni's adjustment, using as weights the enrichment of variant classes with predicted functional impact among association signals estimated from the Icelandic data[91]. This yielded significance thresholds of $1.8 \times 10^{-7}$ for variants with high impact (including stop-gained and loss, frameshift, splice acceptor or donor and initiator codon variants), $3.5 \times 10^{-8}$ for variants with moderate impact (missense, splice-region variants and in-frame indels), $3.2 \times 10^{-9}$ for low-impact variants (including synonymous, 3′- and 5′-UTRs and upstream and downstream variants), $1.6 \times 10^{-9}$ for other variants in DNase I hypersensitivity sites and $5.3 \times 10^{-10}$ for all other variants.

### Variant–variant interaction with liver enzymes

To test whether a specific primary variant is affecting the mean effect of another secondary variant in a given quantitative trait (that is, to test if they are interacting), we split the population into three groups based on the genotype of the primary variant, denoted by $g_p \in \{0, 1, 2\}$. We then estimated the mean effect, $\beta_0, \beta_1$ and $\beta_2$ of the secondary variant

in each group separately, where the quantitative trait was standardized to a normal distribution. We estimated the interaction between the primary and secondary variants by considering the following model:

$$\beta_{g_p} = b + \gamma g_p, g_p \in \{0, 1, 2\},$$

where $b$ and $\gamma$ are the unknown parameters to be estimated. To assess the significance of the interaction parameter $\gamma$, we applied a likelihood ratio test, comparing our model with the null model: $\beta_0 = \beta_1 = \beta_2$ (or equivalently $\gamma = 0$).

### RNA-seq analysis

RNA-seq analysis was performed on whole blood ($n = 17,846$) and subcutaneous adipose tissue ($n = 750$). We isolated RNA using Chemagic Total RNA Kit special (PerkinElmer) in whole blood and RNAzol RT in adipose tissue, according to the manufacturer's protocol (Molecular Research Center, RN 190). The concentration and quality of the RNA were determined using an Agilent 2100 Bioanalyzer (Agilent Technologies). RNA was prepared and sequenced on the Illumina HiSeq 2500 and Illumina Novaseq systems according to the manufacturer's recommendation.

RNA-seq reads were aligned to personalized genomes using the STAR software package v.2.5.3 with Ensembl v.87 gene annotations[92,93]. Gene expression was computed based on personalized transcript abundances estimated using kallisto[94]. Association between sequence variants and gene expression was tested using BOLT-LMM, assuming an additive genetic model and quantile-normalized gene-expression estimates, adjusting for measurements of sequencing artefacts and demography variables. The strongest association within 1 Mb of each gene with $P < 1 \times 10^{-7}$ was called a top cis-eQTL.

Quantification of alternative RNA splicing in whole blood was done using LeafCutter[95]. The cis association between sequence variants and quantified splicing (cis-sQTL) was tested using linear regression assuming an additive genetic model and quantile-normalized percentage-spliced-in values (PSI) of each splice junction, adjusting for measurements of sequencing artefacts, demography variables, and 15 leave-one-chromosome-out principal components of the quantile-normalized PSI matrix. All variants with MAF > 0.2% within 30 kb of each LeafCutter cluster were tested, and the strongest association for each splice junction with $P < 1 \times 10^{-8}$ was called a top cis-sQTL.

### SomaScan proteomic analysis

Plasma samples were collected from Icelanders through two main projects: the Icelandic Cancer Project (52% of participants; samples collected from 2001 to 2005) and various genetic programs at deCODE genetics, Reykjavík, Iceland (48%). The samples collected at deCODE genetics were mainly collected through the population-based deCODE Health study. The average participant age was 55 years (s.d. = 17 years) and 57% were women. In the case of repeated samples for an individual, we selected one of them at random. This left measurements for 39,155 individuals. Of these, 35,559 Icelanders were used in the protein GWASs, because they also had genotype information[50]. All plasma samples were measured with the SomaScan v.4 assay (SomaLogic, Inc.) containing 4,907 aptamers, providing a measurement of relative binding of the plasma sample to each of the aptamers in relative fluorescence units (r.f.u.). When testing for association between protein levels and disease, logistic regression was used with age and sex as covariates. The date of diagnosis was not available and the analysis was therefore not adjusted for the time from diagnosis. A pQTL association is considered to be in cis if the associated variant is located no more than 1 Mb from the transcription start site of the gene that encodes the measured protein, and in trans otherwise. A pQTL was considered to be significant if identified in the previous study ($P < 1.8 \times 10^{-9}$). A top pQTL is the top (most significantly associated) variant per megabase bin.

## Olink proteomic analysis

For a subset of 54,265 individuals in the UKB study (47,151 of British and Irish ancestry), the levels of 1,472 proteins were measured with the Olink Explore 1536 platform as a part of the UKB–Pharma Proteomics Project (UKB application no. 65851). A large majority of the samples were randomly selected across the UKB. The UKB plasma samples were measured using the Olink Explore 1536 platform (https://www.olink.com/products-services/explore) at Olink's facilities in Uppsala, Sweden. Measurements with the Olink Explore platform use the NPX values recommended by the manufacturer, which include normalization. When testing for associations with sequence variants, the protein levels were standardized to a normal distribution.

## Mendelian randomization analysis

We performed Mendelian randomization analyses with the Mendelian-Randomization R package, using both inverse variance, weighted linear regression[96] and Egger regression[97], with default settings.

## Pathway analysis

The *trans* pQTLs that were associated with the identified PDFF variants were grouped together and tested for enrichment in Reactome pathways, Gene ontology terms (biological process, molecular function and cellular component) and PANTHER protein classes. The analysis was performed with the PANTHER v.16.0 tool[98] using Fisher's exact test and false discovery rate (FDR) correction. The associated pQTLs were tested against a reference list that included all measured SomaAmers (*n* = 5,286). Pathway analysis was performed for all variants associating with more than two proteins.

## Liver disease stage classification using plasma proteomics

To examine how effective circulating protein levels (SomaScan and Olink panels) are at discriminating liver disease stages, we trained a logistic regression model with an elastic net penalty on protein levels to classify between individuals diagnosed with cirrhosis and NAFL. To test how this model compares with one trained on liver enzymes, an additional logistic regression model was exclusively trained on age, sex, BMI, ALT, AST and γ-glutamyl transferase. Genetic risk scores for NAFL and cirrhosis from the identified genetic variants were constructed to compare with the protein-based prediction. We created NAFL and cirrhosis GRSs by calculating the sum-of-effect alleles of the NAFLD variants weighted by their cirrhosis and NAFL GWAS effects. In addition, logistic regression models combining the protein scores, liver enzymes and GRSs were trained. In that case, the protein scores were calculated using out-of-fold predictions (with tenfold). The SomaScan analysis was performed on 181 individuals with NAFL and 73 with cirrhosis, and the Olink analysis was performed on 610 individuals with NAFL and 262 with cirrhosis. Models trained to discriminate between the presence of disease diagnosis and population were trained, respectively, on an additional set of 20,619 individuals (SomaScan) and 38,018 individuals (Olink) without cirrhosis and NAFL. The SomaScan measurements were log(transformed) to reduce the effect of outliers. However, we found that this was not necessary for the Olink data. Boruta[99] was used to select relevant features. The selected proteins were preprocessed with Yeo–Johnson power transforms[100] and then scaled to zero mean and unit variance (estimated from the training set) before being fed to the logistic regression model. The elastic net λ and α parameters were tuned using grid search to minimize tenfold crossvalidated average precision. Model performance was evaluated by considering the mean and s.e.m. of the receiver operating characteristic (ROC) area under the curve (AUC) of 1,000 repeated, tenfold-stratified crossvalidation runs.

## Reporting summary

Further information on research design is available in the Nature Research Reporting Summary linked to this article.

## Data availability

GWAS summary statistics for PDFF, NAFL, cirrhosis and HCC are available at https://www.decode.com/summarydata. Sequence variants tested for association have been deposited in the European Variation Archive under accession no. PRJEB15197 (https://www.ebi.ac.uk/ena/browser/view/PRJEB15197). FinnGen data are publicly available and were downloaded from https://finngen.fi. The UKB data were downloaded under application no. 56270. Proteomics data and protein mapping to UniProt identifiers and gene names were provided by SomaLogic and Olink. Other data and code presented in the present study are included in this publication and its Supplementary information.

## Code availability

We used publicly available software together with software/methods developed at deCODE genetics as described in Methods. All customized analysis code is available on request and public software can be found under the following URLs: R (v.3.6.0) and Python extensively used to analyze data and create plots; Graphtyper: https://github.com/Decode-Genetics/graphtyper; multitrait analysis of GWAS summary statistics: https://github.com/JonJala/mtag; PANTHER v.16.0: http://www.pantherdb.org/tools/; Variant Effect Predictor (release 100): https://github.com/Ensembl/ensembl-vep; BOLT-LMM v.2.1: https://data.broadinstitute.org/alkesgroup/BOLT-LMM/downloads; IMPUTE2 v.2.3.1: https://mathgen.stats.ox.ac.uk/impute/impute_v2.html; dbSNP v.140: http://www.ncbi.nlm.nih.gov/SNP/; STAR software package: https://github.com/alexdobin/STAR; Ensembl v.87: https://www.ensembl.org/index.html; LeafCutter v.1: https://github.com/davidaknowles/leafcutter; and kallisto v.0.46: https://github.com/pachterlab/kallisto.

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

## Acknowledgements

We thank all the study participants as well as our colleagues who contributed to data collection, sample handling and genotyping.

## Author contributions

G.S., M.O.U., P.S., D.F.G., H. Holm and K.S. designed the study. G.S., M.O.U., P.S., D.F.G., H. Holm, R.B.T., B.A.J., E.E., G.G., S.R., T.E., E.F., G.H.H., A.H., H. Helgason, G.H., S.H.L., P.M., K.N., A.S.S., G. Thorleifsson, G.L.N., P.S., D.F.G. and K.S. analyzed the data and interpreted the results. D.O.A., M.B., R.G.B., C.E., L.H., D.J., K.U.K., I.O., S.O., S.R.O., O.B.P., E.S., V.S., M.S., G. Thorgeirsson, I.J., H.B., L.N., E.S.B., T.R., U.T., H. Holm and K.S. carried out data collection and subject ascertainment and recruitment. G.S., M.O.U., R.B.T., G.L.N., U.T., P.S., D.F.G., H. Holm and K.S. drafted the manuscript with input and comments from B.A.J., G.R.O., I.C.R. and T.R. All authors contributed to the final version of the manuscript.

## Competing interests

The authors affiliated with deCODE genetics/Amgen, Inc. are employed by the company. The remaining authors declare no competing interests.

## Additional information

**Correspondence and requests for materials** should be addressed to Gardar Sveinbjornsson or Kari Stefansson.

# Reporting Summary

## Statistics

For all statistical analyses, confirm that the following items are present in the figure legend, table legend, main text, or Methods section.

| n/a | Confirmed | |
|---|---|---|
| ☐ | ☒ | The exact sample size ($n$) for each experimental group/condition, given as a discrete number and unit of measurement |
| ☐ | ☒ | A statement on whether measurements were taken from distinct samples or whether the same sample was measured repeatedly |
| ☐ | ☒ | The statistical test(s) used AND whether they are one- or two-sided *Only common tests should be described solely by name; describe more complex techniques in the Methods section.* |
| ☐ | ☒ | A description of all covariates tested |
| ☐ | ☒ | A description of any assumptions or corrections, such as tests of normality and adjustment for multiple comparisons |
| ☐ | ☒ | A full description of the statistical parameters including central tendency (e.g. means) or other basic estimates (e.g. regression coefficient) AND variation (e.g. standard deviation) or associated estimates of uncertainty (e.g. confidence intervals) |
| ☐ | ☒ | For null hypothesis testing, the test statistic (e.g. $F$, $t$, $r$) with confidence intervals, effect sizes, degrees of freedom and $P$ value noted *Give P values as exact values whenever suitable.* |
| ☒ | ☐ | For Bayesian analysis, information on the choice of priors and Markov chain Monte Carlo settings |
| ☒ | ☐ | For hierarchical and complex designs, identification of the appropriate level for tests and full reporting of outcomes |
| ☒ | ☐ | Estimates of effect sizes (e.g. Cohen's $d$, Pearson's $r$), indicating how they were calculated |

*Our web collection on statistics for biologists contains articles on many of the points above.*

## Software and code

Policy information about availability of computer code

| Data collection | No software was used for data collection. |
|---|---|
| Data analysis | We used publicly available software in conjunction with methods developed at deCODE Genetics as described in the methods section.<br><br>Graphtyper version 2, https://github.com/DecodeGenetics/graphtyper;<br>We used R, version 3.6.0 and Python, version 3.9 extensively to analyze data and create plots;<br>Multi-trait analysis of genome-wide association summary statistics version 1, https://github.com/JonJala/mtag;<br>PANTHER v.16.0, http://www.pantherdb.org/tools/;<br>Variant Effect Predictor (release 100), https://github.com/Ensembl/ensembl-vep;<br>BOLT-LMM version 2.1, https://data.broadinstitute.org/alkesgroup/BOLT-LMM/downloads/;<br>IMPUTE2 version 2.3.1, https://mathgen.stats.ox.ac.uk/impute/impute_v2.html;<br>dbSNP version 140, http://www.ncbi.nlm.nih.gov/SNP/;<br>STAR software package, version 2.7.10, https://github.com/alexdobin/STAR;<br>Ensembl version 87, https://www.ensembl.org/index.html;<br>LeafCutter version 1 , https://github.com/davidaknowles/leafcutter;<br>kallisto version 0.46, https://github.com/pachterlab/kallisto |

For manuscripts utilizing custom algorithms or software that are central to the research but not yet described in published literature, software must be made available to editors and reviewers. We strongly encourage code deposition in a community repository (e.g. GitHub). See the Nature Portfolio guidelines for submitting code & software for further information.

## Data

Policy information about availability of data

All manuscripts must include a data availability statement. This statement should provide the following information, where applicable:
- Accession codes, unique identifiers, or web links for publicly available datasets
- A description of any restrictions on data availability
- For clinical datasets or third party data, please ensure that the statement adheres to our policy

GWAS summary statistics for PDFF, NAFL, cirrhosis and HCC are available at https://www.decode.com/summarydata/. Sequence variants tested for association have been deposited in the European Variation Archive under accession number PRJEB15197 (https://www.ebi.ac.uk/ena/browser/view/PRJEB15197). FinnGen data are publicly available and were downloaded from https://finngen.fi/. The UKB data was downloaded under application number 56270. Proteomics data and protein mapping to UniProt identifiers and gene names were provided by SomaLogic and Olink. Other data and code presented in this study are included in this publication and its Supplementary information.

# Field-specific reporting

Please select the one below that is the best fit for your research. If you are not sure, read the appropriate sections before making your selection.

☒ Life sciences ☐ Behavioural & social sciences ☐ Ecological, evolutionary & environmental sciences

For a reference copy of the document with all sections, see nature.com/documents/nr-reporting-summary-flat.pdf

# Life sciences study design

All studies must disclose on these points even when the disclosure is negative.

| | |
|---|---|
| Sample size | Sample sizes are reported in the article and correspond to all available data |
| Data exclusions | No available data was excluded from the study |
| Replication | The NAFLD GWAS analysis was performed using data from 4 populations (Iceland, UK, USA and Finland) and results across populations were compared. |
| Randomization | Not applicatble (GWAS study, not a randomized trial) |
| Blinding | Not applicable (GWAS study, not a randomized trial, so no blinding is required) |

# Reporting for specific materials, systems and methods

We require information from authors about some types of materials, experimental systems and methods used in many studies. Here, indicate whether each material, system or method listed is relevant to your study. If you are not sure if a list item applies to your research, read the appropriate section before selecting a response.

### Materials & experimental systems

| n/a | Involved in the study |
|---|---|
| ☒ | Antibodies |
| ☒ | Eukaryotic cell lines |
| ☒ | Palaeontology and archaeology |
| ☒ | Animals and other organisms |
| ☐ | ☒ Human research participants |
| ☒ | Clinical data |
| ☒ | Dual use research of concern |

### Methods

| n/a | Involved in the study |
|---|---|
| ☒ | ChIP-seq |
| ☒ | Flow cytometry |
| ☒ | MRI-based neuroimaging |

## Human research participants

Policy information about studies involving human research participants

| | |
|---|---|
| Population characteristics | A detailed description of population characteristics can be found in the methods section. The UK Biobank project is a large prospective cohort study of ~500,000 individuals from across the United Kingdom, aged between 40-69 years at recruitment. In the UK Biobank 46% recruited were male, 54% female. 57% aged 40-59 years; 43% aged 60-69 years. The Icelandic deCODE genetics study is based on whole-genome sequence data from 49,708 Icelanders participating in various research projects at deCODE genetics. Variants identified through whole-genome sequencing were imputed into |

155,250 chip-genotyped Icelanders as well as their untyped close relatives based on genealogy.
Finngen summary statistics, including fatty liver disease and cirrhosis, were imported on December 2020 from a source available to researchers (version 4; https://www.finngen.fi/en/access_results) and methods have been documented (https://finngen.gitbook.io/documentation/).
The Copenhagen Hospital Biobank Cardiovascular Study (CHB- CVDC) was used to acquire secondary cardiovascular phenotypes. CHB-CVDC involves a targeted selection of patients with cardiovascular disease from the CHB, a biobank based on patient blood samples drawn in Danish hospitals. For binary phenotypes, the control group included blood donors from The Danish Blood Donor Study (DBDS).
The samples from the US (Intermountain dataset) were WGS using NovaSeq Illumina technology and genotyped using Illumina GSA chips.

| Recruitment | For the deCODE Genetics study individuals were recruited through various research projects at deCODE genetics. The participants are a large fraction of the adult Icelandic population. |
| | UK Biobank holds data on half a million participants throughout the UK. All participants in UK Biobank were recruited through assessment centres, designed specifically for this purpose. |
| | The US data are individuals recruited at the Intermountain healthcare. |
| | The FinnGen database consists of samples collected from Finnish biobanks. |
| | CHB-CVDC involves a targeted selection of patients with cardiovascular disease from the CHB, a biobank based on patient blood samples drawn in Danish hospitals. |

| Ethics oversight | All participating subjects in the deCODE genetics study who donated blood signed informed consent. The personal identities of the participants and biological samples were encrypted by a third-party system. The study was approved by the Icelandic Data Protection Authority and the National Bioethics Committee of Iceland (no VSN-20-182). |
| | The CHB-CVDC has been approved by The National Committee on Health Research Ethics (1708829) and the Danish Data Protection Agency (P-2019-93). The Danish Blood Donor Study (DBDS), approved by the Danish Data Protection Agency (P-2019-99) and the Scientific Ethical Committee system (NVC 1700407). |
| | The FinnGen database consists of samples collected from the Finnish biobanks and and phenotype data collected at the national health registers. The Coordinating Ethics Committee of the Helsinki and Uusimaa Hospital District evaluated and approved the FinnGen research project. The project complies with existing legislation (in particular the Biobank Law and the Personal Data Act). The official data controller of the study is University of Helsinki. |
| | The UK Biobank Resource was used under application number 56270. All phenotype and genotype data were collected following an informed consent obtained from all participants. The North West Research Ethics Committee reviewed and approved UK Biobank's scientific protocol and operational procedures (REC Reference Number: 06/MRE08/65). |
| | For the Intermountain dataset, the Intermountain Healthcare Institutional Review Board approved the study, and all participants provided written informed consent prior to enrollment. |

Note that full information on the approval of the study protocol must also be provided in the manuscript.

