## [Peer Review File · Nature Genetics]

Peer Review Information

Manuscript Title: Multi-omics study of non-alcoholic fatty liver disease

Corresponding author name(s): Dr. Gardar Sveinbjornsson

Reviewer Comments & Decisions:

Decision Letter, initial version:
--

28th January 2022

Dear Gardar,

Your Article "Multi-omics study of non-alcoholic fatty liver disease" has been seen by three referees. You will see from their comments below that, while they find your work of interest, they have raised several relevant points. We are interested in the possibility of publishing your study in Nature Genetics, but we would like to consider your response to these points in the form of a revised manuscript before we make a final decision on publication.

To guide the scope of the revisions, the editors discuss the referee reports in detail within the team, including with the chief editor, with a view to identifying key priorities that should be addressed in revision, and sometimes overruling referee requests that are deemed beyond the scope of the current study. In this case, we think it would be particularly important to assess to what extent the phenotypic classifications include alcohol-related causes of fatty liver and qualify the interpretations accordingly. We also ask that you place the findings in context by citing relevant literature where appropriate and further highlight the findings that are new to this study. We hope you will find this prioritized set of referee points to be useful when revising your study. Please do not hesitate to get in touch if you would like to discuss these issues further.

We therefore invite you to revise your manuscript taking into account all reviewer and editor comments. Please highlight all changes in the manuscript text file. At this stage we will need you to upload a copy of the manuscript in MS Word .docx or similar editable format.

*1) Include a "Response to referees" document detailing, point-by-point, how you addressed each

referee comment. If no action was taken to address a point, you must provide a compelling argument. This response will be sent back to the referees along with the revised manuscript.

*2) If you have not done so already please begin to revise your manuscript so that it conforms to our Article format instructions, available

[here](http://www.nature.com/ng/authors/article_types/index.html).

*3) Include a revised version of any required Reporting Summary:

[REDACTED]

We hope to receive your revised manuscript within 4-8 weeks. If you cannot send it within this time, please let us know.

Nature Genetics is committed to improving transparency in authorship. As part of our efforts in this direction, we are now requesting that all authors identified as 'corresponding author' on published papers create and link their Open Researcher and Contributor Identifier (ORCID) with their account on the Manuscript Tracking System (MTS), prior to acceptance. ORCID helps the scientific community achieve unambiguous attribution of all scholarly contributions. You can create and link your ORCID from the home page of the MTS by clicking on 'Modify my Springer Nature account'. For more information please visit

www.springernature.com/orcid.

Sincerely,
Kyle

2Kyle Vogan, PhD
Senior Editor
Nature Genetics
<https://orcid.org/0000-0001-9565-9665>

Referee expertise:

Referee #1: Genetics, lipids, fatty liver disease

Referee #2: Genetics, gastrointestinal and hepatic disorders

Referee #3: Genetics, metabolic diseases

Reviewers' Comments:

Reviewer #1:
Remarks to the Author:

The work by Stefansson group is a genome-wide association study on NAFL with a clever analyses of nonsense genetic variants to understand the consequences of aminoacidic change into the protein. The work is well written and performed. The main limitation is the classification used to define non-alcoholic fatty liver disease is not specific and in fact it results in the identification among others of a gene involved in ethanol metabolism. Moreover, I do not see ground-breaking novelties in the results presented.

1. The rs11944752 in MTTP is reported as a start of the first codon encoding for methionine. As authors know well MTTP is a pivotal enzyme important for very low density lipoprotein secretion from the liver. Individuals with rare LOF alleles in this gene have hypo/abeta lipoproteinemia and increase in liver fat content. However, in this work the OR reported for a LOF in MTTP NAFLD is protective against NAFLD and this does not make very much sense. Could it be an improper annotation the cause or a switch in the minor and major allele?

2. I would recommend a more accurate citation of the previous GWAS that already have identified the vast majority of the genetic variants presented in this work.

3. For the variant in PNPLA2 I would recommend if it has not been done to perform a sensitivity analyses and adjust for waist/hip ratio and see whether the association with NAFLD holds. Moreover, rare LOF variants cause a neutral storage disease, due to the lack of TGs mobilization from lipid droplets, and has hepatic steatosis as a specific feature <https://www.omim.org/entry/609059?search=pnpla2&highlight=pnpla2>. So the claim the PNPLA2 has never been associated with NAFLD should be revised.

34. There is a large body of evidence showing that the causal variant for the TMC4 locus is on the MBOAT7 gene (PMID: 32591434, PMID: 32253259, PMID: 32058943, PMID: 32859645). I think it is fair to discuss this.
5. As pointed out Emdin et al already found the rs7029757 associated with ALT and cirrhosis. NAFLD is diagnosed clinically by using ALT and cirrhosis is one of the spectra of condition comprising NAFLD. Authors will agree on this reasoning because they continue their work with a GWAS on cirrhosis. Therefore, the claim that rs7029757 is a novel finding is somewhat not true from where I stand and should be amended. Similarly, rs1801689 was already reported by Jamialahmadi et al Gastroenterology 2021
6. The results of the GWAS on cirrhosis are confirmatory of previous findings on GWAS in ALT, see previous comment and the work from Khera's lab and Jamialahmadi et al both in Gastroenterology 2021.
7. Authors claim to have done a GWAS on non-alcoholic fatty liver (NAFL) and among the genome-wide significant find a gene involved in alcohol metabolism (ADH1B). This is a major limitation and suggests that the classification used is not specific for NAFL but overlaps with alcoholic disease.
8. It is mentioned that the variants in TM6SF2 and GPAM had a greater effect in men than women. Has a formal interaction analyses been done to test this?
9. The last part of the paper on the identification of biomarkers is not really linked the rest of the work. The identification of biomarkers requires validation in independent cohorts with longitudinal studies. Moreover, what the authors found is not a biomarker but a signature consisting of 65 and 132 proteins that I do not see the use of in clinical practice. I would remove this part and focus on the human genetics.

Reviewer #2:
Remarks to the Author:

The authors perform a GWAS analysis over a number of liver-related traits in the UK Biobank and combinations with other cohorts for cirrhosis and HCC.

Major points: I find the term "multi-omics study" a bit misleading: this suggests an integrated, novel dataset where multiple dimensions of omics (ideally from the tissue of interest, i.e. liver) are combined and allow therefore particularly new findings. I think this paper is rather an interesting combination of existing datasets. The authors should mention this explicitly and point to the inherent limitations. On a phenotypic side, overlap with alcoholic liver disease is of course a problem.

I would think a focus on the GWAS aspects would point a clearer picture. This is also the strongest part of the study (especially for the liver fat phenotype - where the current list of loci is expanded).

4The serum proteomics adds only mildly to the novelty of the manuscript in my view.

Expression data: I think the authors should focus on liver datasets.

Minor: The TMC4 locus is mostly referred to as MBOAT7 (the neighboring locus in LD) in the literature both for ALD and NAFL.

Reviewer #3:

Remarks to the Author:

Sveinbjornsson, Ulfarsson and colleagues performed several GWAS and multiomic analyses of non-alcoholic fatty liver disease (NAFLD). GWAS for proton density fat fraction in 36K UKBB samples and for NAFLD in ~10K cases & 876K controls from four studies detected 18 association signals at 17 loci. Subsequent GWAS for all-cause cirrhosis and for hepatocellular carcinoma identified 4 signals (2 shared). To interpret genes corresponding to the 20 signals, variants were compared to coding sequences; to top variants for blood, adipose and GTEx expression QTL and splicing QTL; and to rare variant associations. GWAS loci substantially overlapped loci for liver enzyme, lipid, and anthropometric traits. Plasma proteome analyses identified many protein levels associated with NAFLD, as well as pQTL for NAFLD variants, although all as a consequence of disease, not a cause. The strongest predictive models showed that levels of 36 plasma proteins could distinguish NAFLD patients with and without cirrhosis better than models based on commonly measured liver enzyme levels (AUC .92 vs .71), although this prediction is based on only 245 individuals. Additional analyses identified effects of sex and BMI. Together these results further describe genetic contributions to liver disease.

The analysis of NAFL disease status combines data from UKBB, deCODE, FinnGen and Intermountain, and the ~10K cases may provide the largest ever GWAS for disease status. Some analyses test an impressive 46.5 million variants. The subsequent analyses are thorough and make use of large and unique omics datasets. The identification of rare loss-of-function variants in MARC1 and GPAM provides additional insight into whether loss or gain of gene function is associated with NAFLD.

However, novelty of the study is limited. New information in GWAS discovery is largely limited to the distinction between NAFLD disease status and quantitative measures of liver fat or circulating liver enzymes, because the 20 signals have been reported elsewhere for liver quantitative traits. Of five signals described as novel in the manuscript, four have been reported recently in one or more of PMID 34128465, 33972514, 34184762, or 33547301, and the fifth is included in a preprint (www.medrxiv.org/content/10.1101/2021.10.25.21265127v1). In addition, signals detected for liver enzymes have been characterized previously using eQTL, and many of these signals were colocalized with eQTL when reported for lipid or anthropometric traits. At least one previous study also evaluated interactions with BMI (PMID 34184762). The proteomic analysis relevant to NAFLD is novel, although the protein levels largely reflect consequences of disease.

Overall, methods for analyses and quality control choices are appropriate. Significance thresholds

5consider multiple tests appropriately for various analyses.

Major comments:

1. Several genetic analyses of liver enzymes, fatty liver, and related traits have been published in 2021. Results should be compared to those existing publications.

2. Analyses comparing GWAS signals to eQTL, sQTL, and pQTL are based only on linkage disequilibrium between strongest variants. Use of a statistical test to evaluate signal colocalization would be more rigorous.

3. The sQTL results in Table ST6 should be reported with effect allele labels and nucleotide positions for the splice junction tested. For the GUSB and TOR1B variants detected in both the sQTL and eQTL analyses, are the directions of effect on splice products vs transcript levels consistent blood? across tissues?

Minor comments:

1. Figure M2 would be more useful if labels were moved so they do not overlap.

2. GWAS analyses included a surprisingly large number of principal components to adjust for population stratification (e.g. 20, 40), especially for studies limited by country or continental ancestry. A brief rationale could be included in the methods to explain these choices. (Could fewer covariates adjust for stratification sufficiently well and identify more signals?)

Author Rebuttal to Initial comments

Referee expertise:

Referee #1: Genetics, lipids, fatty liver disease

Referee #2: Genetics, gastrointestinal and hepatic disorders

Referee #3: Genetics, metabolic diseases

Reviewers' Comments:

6Reviewer #1:

Remarks to the Author:

The work by Stefansson group is a genome-wide association study on NAFL with a clever analyses of nonsense genetic variants to understand the consequences of aminoacidic change into the protein. The work is well written and performed. The main limitation is the classification used to define non-alcoholic fatty liver disease is not specific and in fact it results in the identification among others of a gene involved in ethanol metabolism. Moreover, I do not see ground-breaking novelties in the results presented.

1. The rs11944752 in MTTP is reported as a start of the first codon encoding for methionine. As authors know well MTTP is a pivotal enzyme important for very low density lipoprotein secretion from the liver. Individuals with rare LOF alleles in this gene have hypo/abeta lipoproteinemia and increase in liver fat content. However, in this work the OR reported for a LOF in MTTP NAFLD is protective against NAFLD and this does not make very much sense. Could it be an improper annotation the cause or a switch in the minor and major allele?

Answer:

The reviewer is correct. We used the transcript NM_001300785.1 for the annotation but the most recent version of the variant effect predictor (VEP) uses NM_001300785.2. rs11944752 is now annotated as an upstream_variant and we have changed the manuscript accordingly.

2. I would recommend a more accurate citation of the previous GWAS that already have identified the vast majority of the genetic variants presented in this work.

Answer:

We have added the following citations to make sure we cite other GWAS studies mentioned by the reviewers.

PMID: 33347879, PMID: 34128465, PMID: 33972514, PMID: 34184762, PMID: 33547301

7We want to emphasize that the most interesting aspect of the study is not the identification of novel NAFLD variants:

- i) We explored the association of identified variants with a spectrum of NAFLD phenotypes, making use of MRI-images (PDF), ICD diagnoses of NAFL, cirrhosis and hepatocellular carcinoma. We also analyzed 51 other phenotypes including liver enzymes, lipids and alcohol dependence. This gives the most comprehensive overview and comparison of the pleiotropic effect of each variant to date.
- ii) This is the first GWAS study that integrates its findings with large proteomics data. Since submission we acquired another large proteomics dataset. Protein levels were measured with 1,459 immunoassays using Olink in 47,151 European participants from the UK Biobank. We integrated our GWAS findings with this dataset and the proteomics section of the manuscript has been updated accordingly. We performed a Mendelian randomization (MR) analysis that shows that levels of many proteins in plasma are altered as a consequence of disease. However, for transferrin receptor protein 1 (TFRC) our data suggest that the transferrin receptor may have a causal role in NAFL. We explored how protein levels can predict disease diagnoses and compared it to predictions using genetic risk scores (GRS) and liver enzymes. The proteome is better at predicting the current stage of disease than a GRS, but a GRS associates with lifetime risk.
- iii) This is the first study to identify a loss-of-function variant in *GPAM* that associates with reduced NAFL risk and we also find a new *MARC1* loss-of-function variant association. This is the first study to observe the high similarity in *GPAM* and *MARC1* variant associations with multiple phenotypic traits.

- iv) This is the first study to demonstrate that an intron variant in *TOR1B* generates a cryptic splice site that elongates exon 2 by 50bp leading to a frameshift that introduces a premature stop codon in exon 3.
- v) This is the first study that explores the longitudinal MRI-PDFF data (measures 2-3 years apart) from the UK Biobank. We demonstrated that PDFF changes with BMI during this time and that there is an interaction between p.Ile148Met and BMI on PDFF in this dataset.
- vi) We describe a method that can be used to estimate MRI-PDFF from raw images.

Reviewer #3 points out that many of the variants identified in the study have been reported in recently published papers, some on liver enzymes or in preprints. Also, reviewer #2 points out that LOFs in *PNPLA2* cause autosomal recessive neutral storage disease with hepatic steatosis as a specific feature and that rs7029757 in *TOR1B* has been associated with cirrhosis and liver enzymes.

The reviewer is correct that the upstream variant to *MTTP* has been reported in a 2021 GWAS on hepatic fat (PMID:34128465).

However, that variants associate with liver enzymes does not mean that they will associate with NAFL. Also, variants can associate with cirrhosis but not NAFL. As pre-prints have not been peer reviewed, we do not consider those. We altered our main text to:

“We identified 18 independent sequence variants at 17 loci in the combined GWAS (Table M2, Table S1), of which four loci have not been reported in a NAFL GWAS”

We also changed the main text: “Novel NAFL variants include a low-frequency” to “The NAFL variants include a low-frequency”

We have also added to the main text:“Homozygous mutations in *PNPLA2* have been associated with neutral lipid storage disease and fatty liver is among its features [REF].”

and we added:

“*APOH* is among the genes that are most highly expressed in the liver^{32,33} and p.Cys325Gly has been associated with measures of liver enzymes [REF]”

3. For the variant in *PNPLA2* I would recommend if it has not been done to perform a sensitivity analyses and adjust for waist/hip ratio and see whether the association with NAFLD holds. Moreover, rare LOF variants cause a neutral storage disease, due to the lack of TGs mobilization from lipid droplets, and has hepatic steatosis as a specific feature <https://www.omim.org/entry/609059?search=pnpla2&highlight=pnpla2>. So the claim the *PNPLA2* has never been associated with NAFLD should be revised.

Answer:

The association of p.Asn252Lys in *PNPLA2* remains significant when adjusting for WHR. We made the following change to the main text:

“The *PNPLA2* p.Asn252Lys has been associated with increased waist-to-hip ratio and HDL-cholesterol levels^{29,30}. Adjusting for WHR does not affect the association (P = 2.4×10^{-10} , effect = 0.20 SD for PDFF).”

See also answer to remark #2

4. There is a large body of evidence showing that the causal variant for the *TMC4* locus is on the *MBOAT7* gene (PMID: 32591434, PMID: 32253259, PMID: 32058943, PMID: 32859645). I think it is fair to discuss this.

Answer:

We agree with the reviewer and have made the following changes to the main text:

10“the missense variant in *TMC4* also associates with liver expression of *MBOAT7*. Loss of hepatic Mboat7 has been shown to associate with NAFLD [Refs]”.

We added references PMID: 32591434, 32253259, 32859645

5. As pointed out Emdin et al already found the rs7029757 associated with ALT and cirrhosis. NAFLD is diagnosed clinically by using ALT and cirrhosis is one of the spectra of condition comprising NAFLD. Authors will agree on this reasoning because they continue their work with a GWAS on cirrhosis. Therefore, the claim that rs7029757 is a novel finding is somewhat not true from where I stand and should be amended. Similarly, rs1801689 was already reported by Jamialahmadi et al Gastroenterology 2021

Answer:

See the answer to remark #2.

Confirming that a variant that associates with liver enzymes also associates with NAFL is important as a variant can associate with cirrhosis risk and alter liver enzymes through other means than NAFL. We feel that this is clear in the current manuscript:

“Rs7029757[A] has been reported to associate with alanine aminotransferase (ALT) levels and cirrhosis but not NAFLD¹³”

We added a reference to Jamialahmadi et al. In that paper rs1801689 is shown to associate with liver enzymes and circulating lipids.

6. The results of the GWAS on cirrhosis are confirmatory of previous findings on GWAS in ALT, see previous comment and the work from Khera’s lab and Jamialahmadi et al both in Gastroenterology 2021.

Answer:

We have referenced these studies.

7. Authors claim to have done a GWAS on non-alcoholic fatty liver (NAFL) and among the genome-wide significant find a gene involved in alcohol metabolism (*ADH1B*). This is a major limitation and suggests that the classification used is not specific for NAFL but overlaps with alcoholic disease.

We performed a GWAS using non-alcoholic fatty liver disease (NAFL) ICD10 code K76.0 diagnoses from Iceland, UK, USA and Finland (N=9,491 cases). These are diagnoses of NAFL from medical doctors. As shown in Table ST1, the variant in *ADH1B* associates with having a NAFL ICD code diagnosis.

The amount of consumed alcohol (2 and 3 units of alcohol daily for women and men respectively) used for distinction of NAFL and alcohol related fatty liver disease (ALD) has been criticized for being arbitrary (PMID: 32974366). In fact, recent evidence suggests that no amount of alcohol consumption is safe for NAFL patients (PMID: 32118004) and it has been argued that NAFL and ALD are strongly related entities and overlap in pathophysiology (PMID: 32974366). ALD diagnosis is only given in the presence of substantial alcohol consumption and it is clear that moderate alcohol consumption (below the ALD diagnostic threshold) can affect the risk of being diagnosed with NAFL.

To assess if the identified variants affect hepatic fat through alcohol consumption, we tested them for association with alcohol dependence using 60,800 cases. Figure M3 and Table S3 show that the *ADH1B* variant is the only one that associates strongly with alcohol dependence. The variant has a minor allele frequency of 3% and carriers of the minor allele consume less alcohol. It should not be surprising that individuals carrying a variant that affects alcohol consumption are less likely to be diagnosed with NAFL since moderate alcohol consumption, less than 2 or 3 units per day, can play a role in accumulation of fat in the liver.

To explore to what extent alcohol liver disease affects our analysis we added an analysis of ICD code diagnoses of alcoholic liver disease using 3,818 cases from Iceland, UK, USA and Finland. We compared the effects of the variants on having a NAFL diagnosis to their effect on having an ALD diagnosis. Since p.His48Arg in

12***ADH1B* affects alcohol consumption, the difference in its effect on ALD and NAFL reflects to what extent alcohol consumption impacts the diagnosis of NAFL. We added to the main text (results section):**

“Among the identified variants, p.His48Arg in *ADH1B* associated with reduced risk of alcohol dependence ($P=2.6 \times 10^{-64}$, $OR=0.31$, $N_{cases}=60,800$). We compared how the variants affect the risk of having a diagnosis of alcohol liver disease (ALD) to the risk of having a NAFL diagnosis (Figure S4). p.His48Arg was the only variant that had a significantly higher effect on risk of ALD ($N_{cases}=3,818$) than of NAFL ($OR=0.33$ and $OR=0.85$, respectively).”

We added the following figure to the Supplementary material:

We added to the discussion:

“p.His48Arg in *ADH1B* associated with less risk of having a NAFL ICD code diagnosis. The variant associates with less alcohol consumption and reduces risk of ALD substantially more than it reduces risk of NAFL. The amount of consumed alcohol (2 and 3 units of alcohol for women and men, respectively) used to distinguish between NAFL and ALD is quite arbitrary [PMID: 32974366]. It is therefore likely that the association of the variant with NAFL is driven by its effect on alcohol consumption below the threshold for ALD diagnosis. None of the other NAFL variants had significantly greater association with risk of ALD than risk of NAFL.”

8. It is mentioned that the variants in *TM6SF2* and *GPAM* had a greater effect in men than women. Has a formal interaction analyses been done to test this?

Answer:

We performed tests of heterogeneity to assess if the effect sizes were significantly different between males and females. P-values are given in the main text.

For clarity we have changed “P-het” to “ $P_{\text{males vs females}}$ ”

“Two out of the 18 variants had greater effect on PDFF in men than women, p.Ile43Val in *GPAM* (effect men=0.09SD, effect women=0.04SD, $P_{\text{males vs females}}=0.00099$) and p.Glu167Lys in *TM6SF2* (effect men=0.40SD, effect women =0.28SD, $P_{\text{males vs females}}=7.3 \times 10^{-5}$).“

9. The last part of the paper on the identification of biomarkers is not really linked the rest of the work. The identification of biomarkers requires validation in independent cohorts with longitudinal studies. Moreover, what the authors found is not a biomarker but a signature consisting of 65 and 132 proteins that I do not see the use of in clinical practice. I would remove this part and focus on the human genetics.

Answer:

To keep a focus on the human genetics and link this analysis to the rest of the work, we added to the analysis a genetic risk score (GRS) based on the identified NAFLD

14variants. We explore if proteomics and genetic risk can discriminate between NAFL and cirrhosis.

We also added an analysis of another large proteomics dataset measured with 1,459 immunoassays using Olink in 47,151 European participants from the UK Biobank.

The results demonstrate that NAFLD can likely be diagnosed and potentially staged with proteomics which requires just a blood draw as opposed to liver biopsy which is the golden standard. The GRS associates with lifetime risk of NAFLD but only adds modest information to the plasma proteome when predicting disease. This is both relevant to clinical practice and relevant to the development of therapeutics. We note that with the emergence of large-scale proteomics platforms such as those provided by Olink and Somalogic a biomarker utilizing tens or hundreds of proteins could soon be clinically useful.

We added the following figure to the main text and moved figure M6 (heatmap of pQTLs) to a Supplementary figure.

The main text now reads:

“To investigate whether plasma protein levels can effectively discriminate between having a NAFL and cirrhosis diagnosis, we trained classification models using liver enzymes, age, sex, and BMI as a baseline as well as using the plasma proteome and genetic risk scores (GRS), using penalized logistic regression. We created NAFL and cirrhosis genetic risk scores (GRS) by calculating the sum of effect alleles of the NAFLD variants weighted by their GWAS effects. In both Iceland (SomaScan) and UK Biobank (Olink) the models trained with the plasma proteome outperformed other

16models in discriminating between NAFL and cirrhosis, NAFL and the population and cirrhosis and the population (Figure M5).”.

Reviewer #2:

Remarks to the Author:

The authors perform a GWAS analysis over a number of liver-related traits in the UK Biobank and combinations with other cohorts for cirrhosis and HCC.

Major points: I find the term "multi-omics study" a bit misleading: this suggests an integrated, novel dataset where multiple dimensions of omics (ideally from the tissue of interest, i.e. liver) are combined and allow therefore particularly new findings. I think this paper is rather an interesting combination of existing datasets. The authors should mention this explicitly and point to the inherent limitations. On a phenotypic side, overlap with alcoholic liver disease is of course a problem.

We do not believe that the term “multi-omics” is misleading. A multi-omics study is a study that combines data from multiple “omes”, such as genome, transcriptome, proteome, metabolome, microbiomes etc. It does not have to involve multiple tissues. This is a large genomic study of NAFLD and the findings are integrated with two large proteomic datasets, our own transcriptomic data, and gene expression data from liver tissues in GTEx. This is therefore a multi-omics study.

With regards to alcoholic liver disease see detailed answer to reviewer #1, comment #7.

I would think a focus on the GWAS aspects would point a clearer picture. This is also the strongest part of the study (especially for the liver fat phenotype - where the current list of loci is expanded). The serum proteomics adds only mildly to the novelty of the manuscript in my view.

See the answer to reviewer 1, remark #9.

17Expression data: I think the authors should focus on liver datasets.

Answer:

We agree with the reviewer that expression data from liver tissue is of high interest. However, large expression datasets from liver tissue with genotype information are difficult to find. The sample size in the genotype-tissue expression (GTEx) project is N=226. We feel that there is no reason not to explore expression datasets from other tissues such as blood, with much larger sample sizes available.

Minor: The TMC4 locus is mostly referred to as MBOAT7 (the neighboring locus in LD) in the literature both for ALD and NAFL.

Answer:

We, agree, see the answer to reviewer #2, remark #4.

Reviewer #3:

Remarks to the Author:

Sveinbjornsson, Ulfarsson and colleagues performed several GWAS and multiomic analyses of non-alcoholic fatty liver disease (NAFLD). GWAS for proton density fat fraction in 36K UKBB samples and for NAFLD in ~10K cases & 876K controls from four studies detected 18 association signals at 17 loci. Subsequent GWAS for all-cause cirrhosis and for hepatocellular carcinoma identified 4 signals (2 shared). To interpret genes corresponding to the 20 signals, variants were compared to coding sequences; to top variants for blood, adipose and GTEx expression QTL and splicing QTL; and to rare variant associations. GWAS loci substantially overlapped loci for liver enzyme, lipid, and anthropometric traits. Plasma proteome analyses identified many protein levels associated with NAFLD, as well as pQTL for NAFLD variants, although all as a consequence of disease, not a cause. The strongest predictive models showed that levels of 36 plasma proteins could distinguish NAFLD patients with and without cirrhosis better than models based on commonly measured liver enzyme levels (AUC .92 vs .71), although this prediction is based on only 245 individuals. Additional analyses identified

18effects of sex and BMI. Together these results further describe genetic contributions to liver disease.

The analysis of NAFL disease status combines data from UKBB, deCODE, FinnGen and Intermountain, and the ~10K cases may provide the largest ever GWAS for disease status. Some analyses test an impressive 46.5 million variants. The subsequent analyses are thorough and make use of large and unique omics datasets. The identification of rare loss-of-function variants in *MARC1* and *GPAM* provides additional insight into whether loss or gain of gene function is associated with NAFLD.

However, novelty of the study is limited. New information in GWAS discovery is largely limited to the distinction between NAFLD disease status and quantitative measures of liver fat or circulating liver enzymes, because the 20 signals have been reported elsewhere for liver quantitative traits. Of five signals described as novel in the manuscript, four have been reported recently in one or more of PMID 34128465, 33972514, 34184762, or 33547301, and the fifth is included in a preprint (www.medrxiv.org/content/10.1101/2021.10.25.21265127v1). In addition, signals detected for liver enzymes have been characterized previously using eQTL, and many of these signals were colocalized with eQTL when reported for lipid or anthropometric traits. At least one previous study also evaluated interactions with BMI (PMID 34184762). The proteomic analysis relevant to NAFLD is novel, although the protein levels largely reflect consequences of disease.

Answer:

With regards to novelty see answer to reviewer #1, remark #2 and we have added the mentioned references to the manuscript.

As the reviewer correctly points out, our Mendelian randomization (MR) analysis demonstrates that protein levels as measured in plasma reflect consequence of disease rather than being causal. This is important to establish and demonstrates that measures of proteins in plasma should be useful in capturing the current stage of the disease.

We have extended the proteomics analysis and it now includes proteomics from the UK Biobank population as measured with Olink. In this dataset we find evidence for a causal role of transferrin receptor protein 1 in NAFLD pathogenesis with the MR analysis. This result is supported by the SomaScan data. The main text now reads:

“The effects of pQTLs of transferrin receptor protein 1 (TFRC) were proportional to their effect on PDFF ($P_{ivw-olink}=5.6 \times 10^{-10}$, $P_{ivw-somascan}=2.0 \times 10^{-4}$, Figure S13, Table S11) suggesting that TFRC may have a causal role in NAFL. Apart from TFRC, the analysis suggests that the alterations in protein level in plasma are not causal, but rather a consequence of disease since for many proteins the effects of the set of NAFL variants on PDFF were proportional to their effects on protein level (Table S12).”

We added a scatter plot to the Supplement showing the effects of TFRC pQTLs on TFRC levels compared to their effect on PDFF.

Since other studies have also explored interactions with BMI, we moved Figure M5 to a Supplementary Figure.

Overall, methods for analyses and quality control choices are appropriate. Significance thresholds consider multiple tests appropriately for various analyses.

Major comments:

1. Several genetic analyses of liver enzymes, fatty liver, and related traits have been published in 2021. Results should be compared to those existing publications.

Answer:

See answer to reviewer #1 comment #2.

2. Analyses comparing GWAS signals to eQTL, sQTL, and pQTL are based only on linkage disequilibrium between strongest variants. Use of a statistical test to evaluate signal colocalization would be more rigorous.

Answer:

20To assess colocalization, more than 16 approaches have been developed recently in addition to the standard method we have applied. Many methods require only summary level data as input rather than exploring LD between variants that associate with disease and the most significant eQTL, sQTL and pQTL. None of these methods are perfect and common challenges include multiple causal variants at the loci, choices of priors, missing causal variants and impact of LD structure.

We have added the r^2 between the disease variant and top-pQTL to Supplementary Table 10 to demonstrate how correlated the disease variants are with the top-pQTLs. This shows that 79% of the identified pQTLs are fully correlated with the disease variant ($r^2 \geq 0.99$) and no other colocalization method would perform better than the r^2 approach (some would perform worse). For 98.3% of the pQTLs the r^2 between the disease variant and the top-pQTLs is > 0.9 demonstrating a high correlation. The SomaScan pQTL analysis has been described in detail in a recently published manuscript in Nature genetics (PMID: 34857953) and summary level data for the Icelandic proteomics GWAS can now be downloaded from <https://www.decode.com/summarydata>. We will also make the NAFLD GWAS data publicly available so any of the numerous co-localization methods can be explored in more detail. The Olink proteomics data will in the future be made publicly available in accordance with UK Biobank policy.

For the sQTLs there is perfect correlation with the disease variant and no other colocalization method would perform better than the r^2 approach. For the eQTL data from gTEX we do not have access to summary level data.

3. The sQTL results in Table ST6 should be reported with effect allele labels and nucleotide positions for the splice junction tested. For the GUSB and TOR1B variants detected in both the sQTL and eQTL analyses, are the directions of effect on splice products vs transcript levels consistent blood? across tissues?

Answer:

For the eQTLs in ST6 we have made the effect directions comparable to the NAFLD risk alleles (as in the main table) and we added *HSD17B13* eQTL data to the table (previous analysis was restricted to NAFL variants). eQTL effect directions are all consistent across tissues.

For *HSD17B13* the sQTLs and eQTLs results are consistent with previous reports.

For the sQTL analysis we have updated Table S6 to include more detailed information, e.g., effect labels and nucleotide positions.

To address the reviewer's question in detail, we added the following text to the Supplementary data:

rs7029757 in *TOR1B* generates a cryptic splice site, elongating exon 2 by 50bp leading to a frameshift and introduces a pre-mature stop codon in exon 3 at position 129807219. Elongates exon 2 by 16 amino acids and introduces a stop codon 10 amino acids into exon 3. The variant is located 2bp upstream of a cryptic splice site. The variant also associates with decreased expression in lungs, pancreas, cell cultured fibroblasts and esophagus mucosa as top-eQTL. Rs7029757 is the top-eQTL for these tissues in the sense that it shows the strongest association at the locus with expression levels of *TOR1B*. For whole blood rs7029757 is not a top-eQTL since another uncorrelated variant associates more strongly with *TOR1B* expression in blood. Rs7029757, however, associates with expression levels in blood after adjusting for stronger associating variants ($-\log_{10}(P)=499.9$, effect = -0.92 SD), decreasing *TOR1B* transcript abundance. This is consistent with the splice effect observed of the canonical splice site (between exon 2 and 3) and eQTL and sQTL results are therefore consistent in whole blood.

For rs6955582 in *GUSB*, the variant is in high LD $r^2=0.99$ with a top-sQTL in whole blood leading to a shift in transcript usage. Rs6955582 does not associate with expression in whole blood.

Minor comments:

1. Figure M2 would be more useful if labels were moved so they do not overlap.

Answer:

We have updated Figure M2.

2. GWAS analyses included a surprisingly large number of principal components to adjust for population stratification (e.g. 20, 40), especially for studies limited by country or continental ancestry. A brief rationale could be included in the methods to explain these choices. (Could fewer covariates adjust for stratification sufficiently well and identify more signals?)

Answer:

For UK Biobank data we used the 40 principal components available from the UK Biobank data showcase and described e.g., in (<https://www.nature.com/articles/s41586-018-0579-z>). The first principal components tag self-reported ethnicity. Since we restrict the analysis to self-reported white ethnicity the first few components should only be important when self-reported ethnicity is wrong. The rest of the principal components are meant to capture population structure within the UK and we feel in general that it is safer to use them to make sure we adjust sufficiently for population structure. However, for the NAFLD phenotypes we see that none of the components associates strongly with the phenotypes and have a minimal impact on the results. The 40 components e.g. explain 0.4% of the variance in MRI-PDFF. Excluding them from the analysis would only minimally affect the GWAS.

Decision Letter, first revision:

Our ref: NG-A59109R

30th April 2022

Dear Gardar,

Your revised manuscript "Multi-omics study of non-alcoholic fatty liver disease" (NG-A59109R) has been seen by the original referees. As you will see from their comments below, they find that the paper has improved in revision, and therefore we will be happy in principle to publish it in Nature Genetics as an Article pending final revisions to comply with our editorial and formatting guidelines.

We are now performing detailed checks on your paper and we will send you a checklist detailing our editorial and formatting requirements soon. Please do not upload the final materials or make any revisions until you receive this additional information from us.

Thank you again for your interest in Nature Genetics. Please do not hesitate to contact me if you have any questions.

Sincerely,
Kyle

24Kyle Vogan, PhD
Senior Editor
Nature Genetics
<https://orcid.org/0000-0001-9565-9665>

Reviewer #1 (Remarks to the Author):

No further comments.

Reviewer #2 (Remarks to the Author):

My concerns regarding the manuscript essentially remain the same. The "multi-Omic" part is the most problematic one. Clearly, there are changes in serum and plasma protein in patients with liver disease - most of those are well known and secondary to the impaired liver function, e.g. iron metabolism changes etc. are known and discussed for a long time.

Reviewer #3 (Remarks to the Author):

This revised manuscript addresses many prior reviewer comments and includes additional references to prior work. The novelty of the study is still somewhat limited for a genetics audience, as a NAFLD audience may best appreciate the characterization of loci with other NAFLD traits, the three loss-of-function variants, and the method and results of MRI-PDFF analyses.

However, this is still the largest GWAS for NAFLD to date, the LOF variants are valuable, and the additional Olink proteomics data on >47,000 UKBB participants provided in revision add support to the SomaScan data and Mendelian randomization results. This evidence that plasma protein levels are often altered as a consequence of disease instead of a cause could have wider implications for other traits, potentially reducing the expectations for GWAS/pQTL colocalization analyses.

Final Decision Letter:

In reply please quote: NG-A59109R1 Sveinbjornsson

2nd September 2022

Dear Gardar,

I am delighted to say that your manuscript "Multi-omics study of non-alcoholic fatty liver disease" has

25been accepted for publication in an upcoming issue of Nature Genetics.

Your paper will be published online after we receive your corrections and will appear in print in the next available issue. You can find out your date of online publication by contacting the Nature Press Office (press@nature.com) after sending your e-proof corrections. Now is the time to inform your Public Relations or Press Office about your paper, as they might be interested in promoting its publication. This will allow them time to prepare an accurate and satisfactory press release. Include your manuscript tracking number (NG-A59109R1) and the name of the journal, which they will need when they contact our Press Office.

Before your paper is published online, we will be distributing a press release to news organizations worldwide, which may very well include details of your work. We are happy for your institution or funding agency to prepare its own press release, but it must mention the embargo date and Nature Genetics. Our Press Office may contact you closer to the time of publication, but if you or your Press Office have any enquiries in the meantime, please contact press@nature.com.

Please note that Nature Genetics is a Transformative Journal (TJ). Authors may publish their research with us through the traditional subscription access route or make their paper immediately open access through payment of an article-processing charge (APC). Authors will not be required to make a final decision about access to their article until it has been accepted. [Find out more about Transformative Journals](https://www.springernature.com/gp/open-research/transformative-journals)

Authors may need to take specific actions to achieve [compliance with funder and institutional open access mandates](https://www.springernature.com/gp/open-research/funding/policy-compliance-faqs). If your research is supported by a funder that requires immediate open access (e.g. according to [Plan S principles](https://www.springernature.com/gp/open-research/plan-s-compliance)), then you should select the gold OA route, and we will direct you to the compliant route where possible. For authors selecting the subscription publication route, the journal's standard licensing terms will need to be accepted, including [self-archiving-and-license-to-publish](https://www.nature.com/nature-portfolio/editorial-policies/self-archiving-and-license-to-publish). Those licensing terms will supersede any other terms that the author or any third party may assert apply to any version of the manuscript.

Please note that Nature Portfolio offers an immediate open access option only for papers that were first submitted after 1 January 2021.

If you have not already done so, we invite you to upload the step-by-step protocols used in this manuscript to the Protocols Exchange, part of our on-line web resource, natureprotocols.com. If you complete the upload by the time you receive your manuscript proofs, we can insert links in your article that lead directly to the protocol details. Your protocol will be made freely available upon publication of your paper. By participating in natureprotocols.com, you are enabling researchers to more readily reproduce or adapt the methodology you use. [Natureprotocols.com](https://natureprotocols.com) is fully searchable, providing your protocols and paper with increased utility and visibility. Please submit your protocol to <https://protocolexchange.researchsquare.com/>. After entering your [nature.com](https://www.nature.com) username and password you will need to enter your manuscript number (NG-A59109R1). Further information can be found at <https://www.nature.com/nature-portfolio/editorial-policies/reporting-standards#protocols>

Sincerely,
Kyle

Kyle Vogan, PhD
Senior Editor
Nature Genetics
<https://orcid.org/0000-0001-9565-9665>